# Using a Low-Cost Sensor to Estimate Fine Particulate Matter: A Case Study in Samutprakarn, Thailand

Supichaya Roddee [1], Supachai Changphuek [1], Supet Jirakajohnkool [1], Panatda Tochaiyaphum [1], Worradorn Phairuang [2,3], Thaneeya Chetiyanukornkul [4] and Yaowatat Boongla [1,*]

1   Department of Sustainable Development Technology, Faculty of Science and Technology, Rangsit Campus, Thammasat University, Pathumtani 12121, Thailand; supichaya.rodd@dome.tu.ac.th (S.R.); supachai.chang@dome.tu.ac.th (S.C.); supet@sci.tu.ac.th (S.J.); catty20@tu.ac.th (P.T.)
2   Faculty of Geosciences and Civil Engineering, Institute of Science and Engineering, Kanazawa University, Kanazawa 920-1192, Japan; pworradorn@se.kanazawa-u.ac.jp
3   Department of Geography, Faculty of Social Sciences, Chiang Mai University, Muang, Chiang Mai 50200, Thailand
4   Department of Biology, Faculty of Science, Chiang Mai University, Muang, Chiang Mai 50200, Thailand; thaneeya.s@cmu.ac.th
*   Correspondence: yaowatat@tu.ac.th; Tel.: +66-924-630-064; Fax: +66-2564-4485

**Abstract:** This study evaluates low-cost sensors (LCSs) for measuring coarse and fine particulate matter (PM) to clarify and measure air pollution. LCSs monitored $PM_{10}$, $PM_{2.5}$ (fine particulates), and $PM_{1.0}$ concentrations at four sites in Samutprakarn, Thailand from December 2021 to April 2022. Average daily $PM_{10}$, $PM_{2.5}$, and $PM_{1.0}$ concentrations at the monitoring locations were 53–79, 34–45, and 31–43 µg/m$^3$, respectively. In December 2021, the monitoring station had a daily $PM_{2.5}$ value above 100 µg/m$^3$, indicating haze occurrences. However, the monitoring site's daily $PM_{10}$ and $PM_{1.0}$ concentrations did not surpass Thailand's ambient air quality threshold. We also measured and calibrated comparative particulate matter concentrations from LCSs and a tapered element oscillating microbalance (TEOM) monitor (Pollution Control Department (PCD) standard analytical method). $PM_{2.5}$ concentrations from the LCSs were lower than TEOM, but the difference was not statistically significant. The $PM_{2.5}$ monitoring station provided near-real-time air quality data for health risk reduction, especially when PM levels were high. Based on this study, authorities and local agencies may consider improving air quality regulation in Samutprakan, focusing on suburban $PM_{2.5}$ air pollution.

**Keywords:** fine particles; low-cost sensor; PM sensor; Samutprakan; Thailand

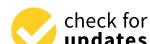



## 1. Introduction

Particulate matter (PM) refers to a complex mixture of small solid particles and liquid droplets that are suspended in the air and vary in size, shape, surface area, chemical composition, solubility, and origin. Particles in the air with an aerodynamic diameter up to 10 µm are labeled as $PM_{10}$, while particles in the air with an aerodynamic diameter between 2.5 and 10 µm are referred to as $PM_{10-2.5}$ or coarse particles. Additionally, particles with aerodynamic diameters up to 2.5 µm are termed fine particle matter ($PM_{2.5}$). Ultrafine particles with a diameter below 1 micron are called $PM_{1.0}$ [1–3]. Both anthropogenic (man-made) and natural PM are known [2,3]. Several sources have shown that $PM_{10}$ sources include industrial activities, transportation, fuel combustion, and construction [4–6], and $PM_{2.5}$ sources include traffic/vehicular [7,8], residential wood burnings, and forest fires [9–11]. Hazardous health effects from PM have been extensively studied [12,13]. Epidemiological evidence has been provided for the human respiratory system [13–15]. Due to their size [16,17], short-term and long-term exposures [18,19] of ambient $PM_{2.5}$ cause more adverse health effects and toxicity than $PM_{10}$ [20–22]. Ambient $PM_{2.5}$ can easily move through

the human body inhalation pathway. Before the COVID-19 lockdown, many countries were considered high-risk areas and needed to address the human health effects of the $PM_{2.5}$ concentration distribution. During and after the COVID-19 lockdown, the concept was to reduce air quality (AQ) due to the strict control of the virus transmission, and this potentially led to an air quality change in high-risk areas [23–25]. In addition, an improvement in air quality resulted from policies that controlled and monitored ambient air (AA) monitoring [26–29]. Thailand has serious air quality problems in several areas in every dry season (haze episodes from December to February). Excessive average levels of particulate matter ($PM_{2.5} > 50$ μg/m$^3$ is the ambient air standard value in Thailand) are found in the air in many areas of Bangkok [30–33] and Thailand [34,35]. To assess the harmful health impact of ambient air pollution, the air quality index equation was proposed for calculating AQI values. The Thailand Air Quality Index is based on five criteria and each color has meaning in terms of the air pollution level. The index range is from 0 to above 201. The AQI range from 0 to 25 is considered very good air quality and appropriate for outdoor activities and tourism (Excellent: color blue). The AQI range from 26 to 51 is considered good air quality, and outdoor activities and tourism are possible (Satisfactory: color green). For the AQI range from 51 to 100 (Moderate: color yellow), the general public is able to engage in outdoor activities, but sensitive groups are recommended to undertake limited outdoor activity if symptoms such as coughing, difficulty in breathing, and/or eye irritation occur. In the AQI range from 101 to 200 (Very Unhealthy: color orange), the general public should monitor their health. If symptoms such as coughing, difficult breathing, and/or eye irritation occur, outdoor activities should be limited and/or personal protective equipment should be used as needed. For sensitive groups, reduced/minimized outdoor activities and/or use of personal protective equipment is recommended. If symptoms such as coughing, difficulty in breathing, eye irritation, chest pains, headaches, irregular heartbeat, nausea and/or exhaustion occur, medical assistance should be sought. If AQI values are above 201 (Hazardous: color red), outdoor activities should be avoided by all. Additionally, areas with poor air quality should be avoided and/or personal protective equipment should be used as needed. If any symptoms occur, medical assistance should be sought.

The air quality monitoring station in Thailand was provided by the Pollution Control Department (PCD). The PCD has created air quality monitoring networks to measure air pollutants such as $PM_{10}$, $PM_{2.5}$, and noxious gases (CO, $NO_2$, $SO_2$, and $O_3$). Samutprakarn province is a city near Bangkok (BKK), Thailand. Samutprakarn is an area that has mixed air emission sources, associated with many types of surrounding industry, fuel combustion, high population density, and vehicular sources [36]. The PCD has provided reference equipment [37–39] to measure air pollutants in Thailand, including the gravimetric method for $PM_{10}$ and the federal equivalent method (FEM) for $PM_{2.5}$, as well as the tapered element oscillating microbalance (TEOM) [40–42] and the beta attenuation monitor (BAM). However, these instruments are quite expensive, and they are only deployed at central monitoring sites in each province. One monitoring station cannot record the spatial distribution of PM concentrations. PCD started to monitor ambient air in Samutprakarn province with TEOM. Authorities and local agencies of Samutprakarn province have developed policies for improving air quality in the Samutprakarn region, and air monitoring instruments are used during haze seasons and in the non-haze seasons. However, such instruments could not be distributed to all areas due to the high cost and need for trained personnel. Currently, air pollutants such as $PM_{10}$ and $PM_{2.5}$ are instead measured by several low-cost instruments and equipment.

LCSs have been used extensively to characterize air pollution in the last few years. LCSs have been widely used in monitoring $PM_{10}$ and $PM_{2.5}$ mass concentrations. In general, the LCS is a device that uses one or more sensors with other components to detect, monitor and report on specific air pollutants, such as particulate matter (PM), carbon dioxide ($CO_2$), and/or environmental factors like temperature and RH. Several reports in Thailand and in other countries have reported $PM_{10}$ and $PM_{2.5}$ levels from many sensor

devices [43–46]. Calibration of LCSs is critical due to aerosols' physical and chemical properties and weather conditions. Calibration of the LCS in the field typically involves comparing it to a reference instrument. Chunitiphisan et al. [44] measured particle matter, temperature, and RH with an Unmanned Aerial Vehicle (UAV) and a sensor (Plantower PMS 3003) in Northern Thailand (Nan province). Both $PM_{10}$ and $PM_{2.5}$ levels were found to be lower than the data from reference devices (TEOM); measured $PM_{10}$ values were 10.25 $\mu g/m^3$ using the Plantower PMS 3003 (light scattering technique) and 22.69 $\mu g/m^3$ using the TEOM, and $PM_{2.5}$ was 8.84 using the Plantower PMS3003 and 11.84 $\mu g/m^3$ using the TEOM. The linear correlation of the particulate matter between their sensor and reference devices was 0.5 ($PM_{10}$) and 0.6 ($PM_{2.5}$). Moreover, a light scattering technique was used to measure $PM_{10}$ and $PM_{2.5}$ outdoors in an area of haze pollution in Chiangmai province, Thailand. During the haze episode, $PM_{10}$ and $PM_{2.5}$ were collected in several areas around Chiangmai over a twenty-four-hour period, and the results were between 12.58 and 149.82 $\mu g/m^3$ ($PM_{10}$) and from 9.63 to 130.37 $\mu g/m^3$ ($PM_{2.5}$), respectively. In this study, beta ray detectors were employed to estimate the correlation of $PM_{10}$ and $PM_{2.5}$ levels in their study versus a certified device (TEOM). These observations were in agreement with results in the reported reference device for $PM_{10}$ and $PM_{2.5}$ (r = 0.8658). Thus, their sensor has the potential for detecting $PM_{10}$ and $PM_{2.5}$ [45]. Another application of the $PM_{2.5}$ concentration sensor (SN-GCHA1, Panasonic Photo & Lighting Co., Ltd., Osaka, Japan), the light scattering technique, revealed the mass concentration results from real-time indoor measurements (ranging from 20.05–45.85 $\mu g/m^3$) and outdoor measurements (ranging from 9.42–56.56 $\mu g/m^3$). Additionally, data were also collected for the hourly average temperature (28.8 to 35.2 °C) (temperature sensor: RHHI-112A, Shinyei Technology Co, Ltd., Kobe, Japan) and RH (55.2 to 72.6%) (RH sensor: RHHI-112A, Shinyei Technology Co, Ltd.). In a controlled laboratory environment, Levy Zamora et al. [47] tested three Plantower PMSA003 sensors against eight emission sources: incense oleic acid, NaCl, talcum powder, cooking emissions, and monodispersed polystyrene latex spheres. The accuracy ranged from 13% to more than 90% when compared to reference instruments, demonstrating that PM sources had an impact on LCS performance. The LCS was most accurate for sizes under 1 $\mu m$. Kim et al. [48] evaluated LCSs in the laboratory and in the field using two commercial LCSs, Plantower PMS3003 and Plantower PMS7003, which were compared with a reference-grade PM monitor (GRIMM 11-D). The LCSs indicated lower mass concentrations than GRIMM 11-D for laboratory testing while the LCSs showed generally higher PM mass concentrations than GRIMM 11-D in field testing. The result indicated that the outdoor environment had notable impacts on LCS data. Field testing revealed that the LCS error increased with increasing RH levels (>75%). Additionally, meteorological conditions, including wind speed, wind direction, and topographic elevation [49] had an effect on PM concentrations, while combustion emission sources played a significant part in the $PM_{10}$ and $PM_{2.5}$ concentrations. Levy Zamora et al. [47] indicated that temperature and RH significantly influenced LCS performance [47]. Long-term monitoring of $PM_{2.5}$ levels in Hanoi was performed using Panasonic $PM_{2.5}$ sensors (light scattering intensities of single particles technique), and the sensors were calibrated with virtually monodisperse polystyrene latex (PSL) particles. The sensor detects particles with diameters as small as ~0.3 $\mu m$ and measures $PM_{2.5}$ mass concentrations up to ~600 $\mu g/m^3$. The sensors correlated well with a nearby beta attenuation monitor (3.1 km away), with an $R^2$ value of 0.73 [50]. Ly et al. [51] employed a machine learning technique based on a random forest (RF) algorithm and concentration weight trajectory (CWT) to investigate the impact of long-range transport and meteorological data on $PM_{2.5}$ characteristics and LCS (light scattering intensities of single particles technique) performance. The influence of meteorological and regional transport on winter $PM_{2.5}$ levels was also investigated. In an urban city of Klang Valley, in Malaysia, the $PM_{2.5}$ level was found to be 19.1 $\mu g/m^3$ using a detector (AiRBOXSense, Alphasense Ltd., Great Notley, Braintree, UK, the optical particle sensor (OPC-N2)). A GRIMM portable aerosol spectrometer (PAS-1.108) was employed as a reference device to validate $PM_{2.5}$ and $PM_{10}$ levels, with correlation coefficients of 0.71

and 0.83, respectively [52]. The LCS, namely LCS Edimax AirBox AI-1001W V3 (laser sensor), was used to measure the $PM_{2.5}$ levels in indoor and outdoor areas in West Jakarta, Indonesia. This sensor was connected to the Internet for 24 h. The sensor specifications were as follows: concentration range of 0–500 µg/m$^3$; smallest particle size of 0.3 µg; maximum weight of 210 g; sensor temperature of 0–60 °C $\pm$ 1 °C; relative humidity (RH) range of 0–100% $\pm$ 5% RH; operation temperature range of $-10$–60 °C; with one internal antenna. The LCSs were placed at a height of 2.5–3 m, which corresponded to the air quality monitoring station (AQMS) of $PM_{2.5}$. According to this study, $PM_{2.5}$ concentrations are higher outdoors than indoors [53]. A Pocket $PM_{2.5}$ Sensor (LED: light-emitting diode, PD: photodiode, USB: Universal Serial Bus) (Yaguchi Electric Co., Ltd., Miyagi, Japan) was used to measure the $PM_{2.5}$ concentrations in seven townships in Yangon, Myanmar. The Pocket $PM_{2.5}$ sensor showed a strong correlation with the reference monitor [2]. One of the townships showed a high level of $PM_{2.5}$, over 100 µg/m$^3$. This result indicated that the particular monitoring site had an effect on the $PM_{2.5}$ levels. Also, solar radiation research aids climate models and meteorological forecasts. Meteorological factors, including dawn duration, temperature, and RH, have been used to assess solar radiation at various places. The study showed that higher levels of air pollution were associated with increased solar radiation, which was the greatest during the summer [54–56]. PM concentration using a LCS as a detector showed positive $R^2$ with a reference device [57,58]. Calibration under actual ambient conditions is necessary before deployment for monitoring $PM_{2.5}$ levels. The LCS should be calibrated for each source in their appropriate environment [58,59].

The LCS has lower accuracy and sensitivity than the reference monitor, and its use is dependent on the type of sensor. However, given their low cost, portability, and ease of installation, LCSs are suited for areas where reference monitors are not established. As a result, we assessed the performance of an LCS (PMS7003, Plantower) to better understand the major elements influencing performance, such as urban emission sources, RH, temperature, and $PM_{10}$, $PM_{2.5}$, and $PM_{1.0}$ levels. Furthermore, data storage for measurements is vital; we examine wire-free data export to an application, giving quick access to the information. The objective of our study was to develop a LCS device for measuring coarse ($PM_{10}$) and fine particulate matter ($PM_{2.5}$ and $PM_{1.0}$), along with some environmental data including air temperature and RH. The LCS was calibrated with reference devices from Thailand. The field test was performed at our monitoring sites, which were known to have serious air quality problems. The study location was established in Samutprakarn province, which is an urban environment area. Between December 2021 and April 2022, we collected mass concentrations of $PM_{10}$, $PM_{2.5}$, and $PM_{1.0}$, as well as temperature and RH data. The tested LCS was operating for a period of five months, The PM results were converted to AQI values using Thailand's AQI equation and then shown as geographic information system (GIS) maps.

## 2. Materials and Methods

### 2.1. Monitoring Sites

The monitoring stations are in Thailand's Samutprakarn province, a high-risk location for air pollution. Samutprakarn is a 388 square-kilometer area with a warm temperate, semi-humid continental monsoon climate with two seasons in terms of temperature and RH. The yearly average temperature is 28.8–32.1 °C, and the annual rainfall is 1100–1500 mm. The resident population is 1,344,875 million people [60]. Figure 1A–F shows the installation of four LCSs at four sampling sites for this study. Each sampling site had different characteristics: the first site was in a residential area (DFH1, Latitude, 13.607976, Longitude, 100.744788, Figure 1C); the second site was located in a semi-residential area in the Bangpli district, Samutprakarn province, surrounded by a mixture of many combustion emission sources (cooking, vehicles, construction) (DFH2, Latitude, 13.601481, Longitude, 100.770638, Figure 1D); the third site (DFH3, Latitude, 13.596204, Longitude, 100.639773, Figure 1E) was a closed to residential area in Pak Nam, Meuang, Samutprakarn province; and the last site (DFH4, Latitude, 13.571715, Longitude, 100.786608, Figure 1F) was set up near a

commercial area in the Bang Sao Thong district, Samutprakarn province, that had twenty-four-hour a day transportation (Table 1). A Pollution Control Department (PCD)-approved instrument collected daily $PM_{10}$ and $PM_{2.5}$ readings for comparison with our LCSs. The PCD stations measured the air quality in Pak Nam, Meuang, Samutprakarn province (18T, residential area, Figures 1A and S1); this was near the monitoring location of our LCS (DFH3) and in Bang Sao Thong district, Samutprakarn province (19T, semi-residential area, Figures 1B and S2); this was near the monitoring location of our LCS (DFH4). A Tapered Element Oscillating Microbalance (TEOM) (Table S2) was used to gather ambient air samples from the 18T and 19T stations [61]. The $PM_{10}$ and $PM_{2.5}$ mass concentration data were completely delivered to these two PCD stations in Samutprakarn. The LCSs have been placed >1.5 m. The distances between 18T and DFH1 were 16.09 km, DFH2 18.89 km, DFH3 4.70 km, and DFH4 20.90 km. Additionally, the distances between 19T and DFH1 were 6.05 km, DFH2 3.72 km, DFH3 16.12 km, and DFH4 0.13 km. The list of acronyms used in study was presented in Table S1.

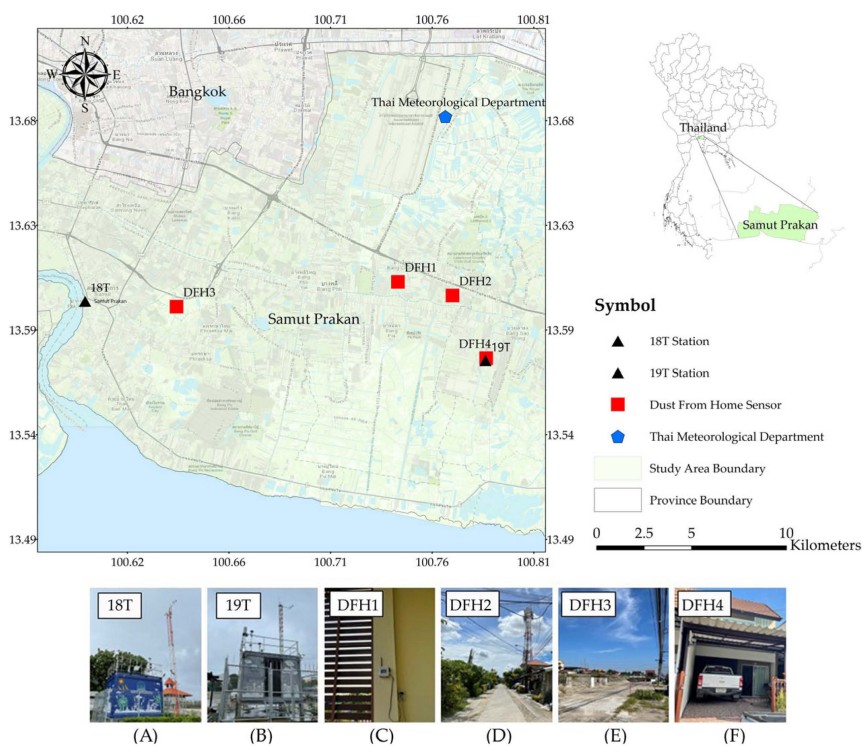

**Figure 1.** The selected monitoring sites are in Samutprakarn province, Thailand.

**Table 1.** Monitoring sites of real-time $PM_{10}$, $PM_{2.5}$, and $PM_{1.0}$ concentrations (24 h).

| Monitoring Site Code | Specify of Site | Parameter | Instrument |
|---|---|---|---|
| DFH1 | Residential area | $PM_{10}$, $PM_{2.5}$, $PM_{1.0}$, RH, Temp. | LCS |
| DFH2 | Semi-residential area | $PM_{10}$, $PM_{2.5}$, $PM_{1.0}$, RH, Temp. | LCS |
| DFH3 | Residential | $PM_{10}$, $PM_{2.5}$, $PM_{1.0}$, RH, Temp. | LCS |
| DFH4 | Commercial area | $PM_{10}$, $PM_{2.5}$, $PM_{1.0}$, RH, Temp. | LCS |
| PCD station (18T) | Residential area | $PM_{10}$, $PM_{2.5}$ | TEOM |
| PCD station (19T) | Semi-residential area | $PM_{10}$, $PM_{2.5}$ | TEOM |

### 2.2. Details of Sensor Device

In this study, a LCS (Plantower Laser Dust Sensor PMS7003) was tested at several monitoring sites, from December 2021 to April 2022. The Plantower Laser Dust Sensor PMS7003 is a low-cost, commercially available sensor that costs between THB 800–1100 (USD 25–35). The advantages of this sensor are that it can be controlled remotely via a

Raspberry Pi and is small enough to fit into a mobile or wearable enclosure, while previous research has shown that, compared with other sensors such as Plantower PMS5003 and Alphasense OPC-N2, PMS7003 tends to show a significant correlation with the reference instrument and good reproducibility [47,48,62]. Therefore, our study employed PMS7003. To assess the performance of our LCS for the detection of the $PM_{10}$, $PM_{2.5}$, and $PM_{1.0}$ concentrations, the RH and temperature, and various test data, were recorded. All the devices used were obtained from an electronics store and collocated; they included the Plantower laser dust sensor PMS7003 (a particle detector that uses the light scattering principle), a NodeMCU ESP8266 (V2) Wifi board, a fan (DC 12V, 8 cm), a liquid crystal monitor display (I2C LCD2004A LCD (Blue Screen), a 20 $\times$ 4 LCD with back-light), a power switching supply (DC 12V, 12 $\times$ 12 mm), a USB adapter (5V, 1A), a micro USB (B) programming cable (30 cm), a digital temperature and RH sensor (DHT22 AM3202), plastic cover boxes, and a jumper (F2F). The data gathered were reported via the Internet of Things (IoT) and was periodically uploaded to remote cloud storage such as Thingspeak. A numerical and graphical value near the real-time monitor was automatically obtained.

### 2.3. Temperature and RH Sensor

The impact of meteorological conditions on the $PM_{10}$, $PM_{2.5}$, and $PM_{1.0}$ levels was evaluated using the LCSs and data from Thailand's Meteorological Department (TMD) [63]. TMD station is shown in Figure 1. The distances between TMD and DFH1 were 9.04 km, DFH2 9.43 km, DFH3 17.04 km, and DFH4 12.89 km. The LCS (DHT22, AM3202) measured the environmental parameters, such as air temperature ($^{\circ}$C) and RH (%). The air temperature monitoring sensor had an air temperature range of 0 $^{\circ}$C to +60 $^{\circ}$C (accuracy $\pm$ 0.2 $^{\circ}$C, resolution 0.1 $^{\circ}$C), and the RH monitoring sensor had a range of 0 to 60% (accuracy $\pm$ 2%, resolution 0.1%). Four LCSs were placed at different locations (residential, semi-residential, and commercial areas) in various environments as mentioned in Table 1. These four locations had varying meteorological conditions and pollution levels. However, no differences were observed in the temperature and RH values between the TMD and our LCSs over the monitoring period.

### 2.4. PM Mass Concentration

The LCS (Plantower Laser Dust Sensor PMS7003) was used to measure the concentrations of $PM_{10}$, $PM_{2.5}$, and $PM_{1.0}$, which are widely found in urban environments. The concentration range was 1–999.90 $\mu$g/m$^3$. This LCS is a digital and universal particle concentration sensor that can be used to determine the number of particles present in the surrounding air. For this sensor, the laser scattering principle is used (irradiation of particles in ambient air), then light scattered to a specific angle is collected, and lastly the scattered light signal change with time curve is obtained. Finally, the microprocessor is used to determine the equivalent particle diameter and the number of particles with varied sizes per unit volume.

### 2.5. Air Quality Index (AQI)

The air quality index (AQI) was utilized in accordance with Thailand's air quality index standard, as released by the PCD [61]. The Thai government uses the AQI value to communicate the quality of the air to the general public. Additionally, the AQI is divided into five categories, ranging from 0 to 201, with each category employing colors to symbolize the level of danger. The air pollutants include $PM_{10}$, $PM_{2.5}$, $O_3$, $SO_2$, NOx, and CO. Each of these contaminants has its own set of air quality standards that are utilized to calculate the overall AQI. Equation (1) is used to calculate pollutant levels [61]. The total AQI values were then entered into geographical information systems (GIS) and plotted on a map. The geographical interpolation of the AQI is demonstrated using GIS. The first equation is as follows:

$$I = \frac{Ij - Ii}{Xj - Xi} (X - Xi) + Ii \tag{1}$$

where $I$ = air quality index (AQI) for pollutant; $X$ = pollutant concentration; $Xi$ = the concentration breakpoint that is $\leq X$; $Xj$ = the concentration breakpoint that is $\geq X$; $Ii$ = the index breakpoint corresponding to $Ii$; $Ij$ = the index breakpoint corresponding to $Ij$.

*2.6. Statistical Analysis*

This experiment was completed using the Statistical Package for Social Science (SPSS) software version 28, and a *t*-test (*t*-test used to test difference between the means of two variables) was utilized to detect the difference between the concentrations from the LCSs and TEOM. The average, minimum, and maximum values, standard error, mean difference, median, standard deviation, variance, and observation results were generated (Table 2). A one-way analysis of variance (ANOVA) and a post-hoc test were used to compare the concentrations of $PM_{10}$, $PM_{2.5}$, and $PM_{1.0}$ at each monitoring site. A statistically significant level of *p*-value = 0.05 was used to determine the dust, temperature, and RH levels at each LCS monitor site and whether there are statistical differences between each sampling point.

**Table 2.** PM Concentrations of each sensor in the study location.

| Parameters | Particle Size-Range | | |
| --- | --- | --- | --- |
| | $PM_{10}$ $\mu g/m^3$ | $PM_{2.5}$ $\mu g/m^3$ | $PM_{1.0}$ $\mu g/m^3$ |
| DFH-1 | 59 | 41 | 33 |
| DFH-2 | 58 | 39 | 32 |
| DFH-3 | 60 | 40 | 34 |
| DFH-4 | 52 | 31 | 30 |
| PCD (18T) | 62 | 38 | NA |
| PCD (19T) | 56 | 31 | NA |
| Mean | 58 | 36 | 33 |
| S.D. | 3 | 4 | 2 |
| Max | 62 | 41 | 34 |
| Min | 52 | 31 | 30 |
| *t*-stat | | | |
| DFH-1 vs. 18T | −0.9 | 1.6 | NA * |
| DFH-2 vs. 18T | 1.5 [a] | −0.5 | NA * |
| DFH-3 vs. 18T | 0.5 | −0.9 | NA * |
| DFH-4 vs. 18T | 3.3 | 3.3 | NA * |
| DFH-1 vs. 19T | −1.4 | −6.1 | NA * |
| DFH-2 vs. 19T | −0.8 | −0.5 | NA * |
| DFH-3 vs. 19T | −1.5 | −4.8 | NA * |
| DFH-4 vs. 19T | 1.5 | −0.1 [a] | NA * |

* NA: not available, [a] statistical analysis with significant difference level at 0.05.

*2.7. Geographic Information System (GIS)*

In this study, ArcMap version 10.8 software was used and employed the clip tool to make a map that indicates where the measurement instrument is. The buffer command was used to manipulate map data for the study area. The data were then displayed using the colors based on the Thai AQI Index, and the coordinates of the PM measuring devices were specified for display purposes. Reference devices of the Pollution Control Department at two stations were used for comparison data. For better visual comprehension, the findings are arranged geographically by color and Thai AQI value based on severity.

**3. Result and Discussion**

*3.1. Calibration and Validation of Sensor*

The combination of LCS enables measurement of a range of parameters such as temperature (°C), RH (%), and coarse and fine particle distributions with the Plantower Laser Dust Sensor PMS7003, as shown in Figure 2a,b. The LCS array for each monitoring site (DFH1-DFH4) was tested, and the PM concentrations were collected before starting the long-term monitoring. The particulate matter (PM) sensors were calibrated against

an instrument approved by the PCD Station at Samutprakarn [3]. The LCSs for $PM_{10}$, $PM_{2.5}$, and $PM_{1.0}$ were set up at the monitoring sites, and their results were compared with a TEOM monitor (18T, 19T). The PM concentrations from the LCSs and TEOM are shown in Figure 3. The highest mass concentration in all LCSs was $PM_{10}$, followed by $PM_{2.5}$ and $PM_{1.0}$, respectively. Figure 3a shows that the DFH1 sensor recorded an average $PM_{10}$ concentration of 60 $\mu g/m^3$ and $PM_{2.5}$ concentrations of 41 $\mu g/m^3$ (Figure 3b), which were lower than the mass concentrations of 18T ($PM_{10}$ = 79, $PM_{2.5}$ = 46 $\mu g/m^3$) and 19T ($PM_{10}$ = 63, $PM_{2.5}$ = 36 $\mu g/m^3$). The LCS DFH2 had a lower average concentration than the TEOM monitor (18T, 19T). The average mass concentration of $PM_{10}$ was 54 $\mu g/m^3$ and for $PM_{2.5}$ it was 34 $\mu g/m^3$. The average mass concentration of $PM_{10}$ with DFH3 was similar to 18T (79 $\mu g/m^3$), but 19T showed a low concentration. The average $PM_{2.5}$ mass concentrations were 45 $\mu g/m^3$ (DFH3), 46 $\mu g/m^3$ (18T), and 36 $\mu g/m^3$ (19T). DFH4 had an average $PM_{10}$ mass concentration of 62 $\mu g/m^3$, followed by 36 $\mu g/m^3$ for $PM_{2.5}$. The measurement value of both reference devices (18T, 19T) was greater than the LCS in DFH4.

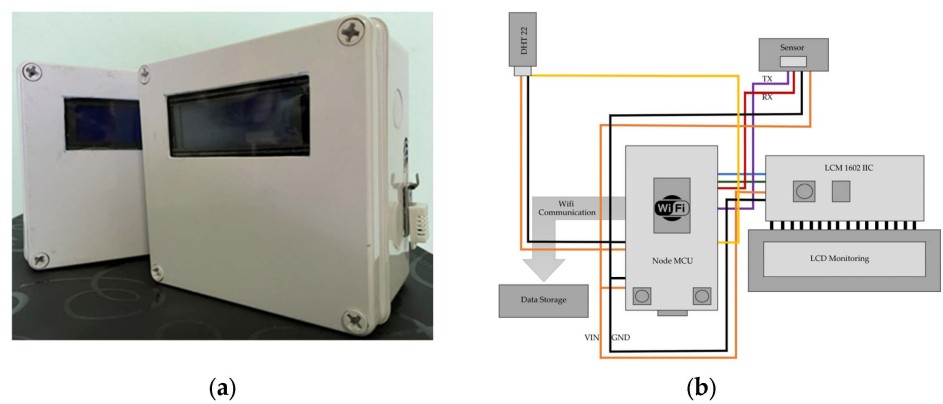

(**a**)                                                                                   (**b**)

**Figure 2.** The LCS used in this study (PMS7003), (**a**) outside box, (**b**) LCS diagram.

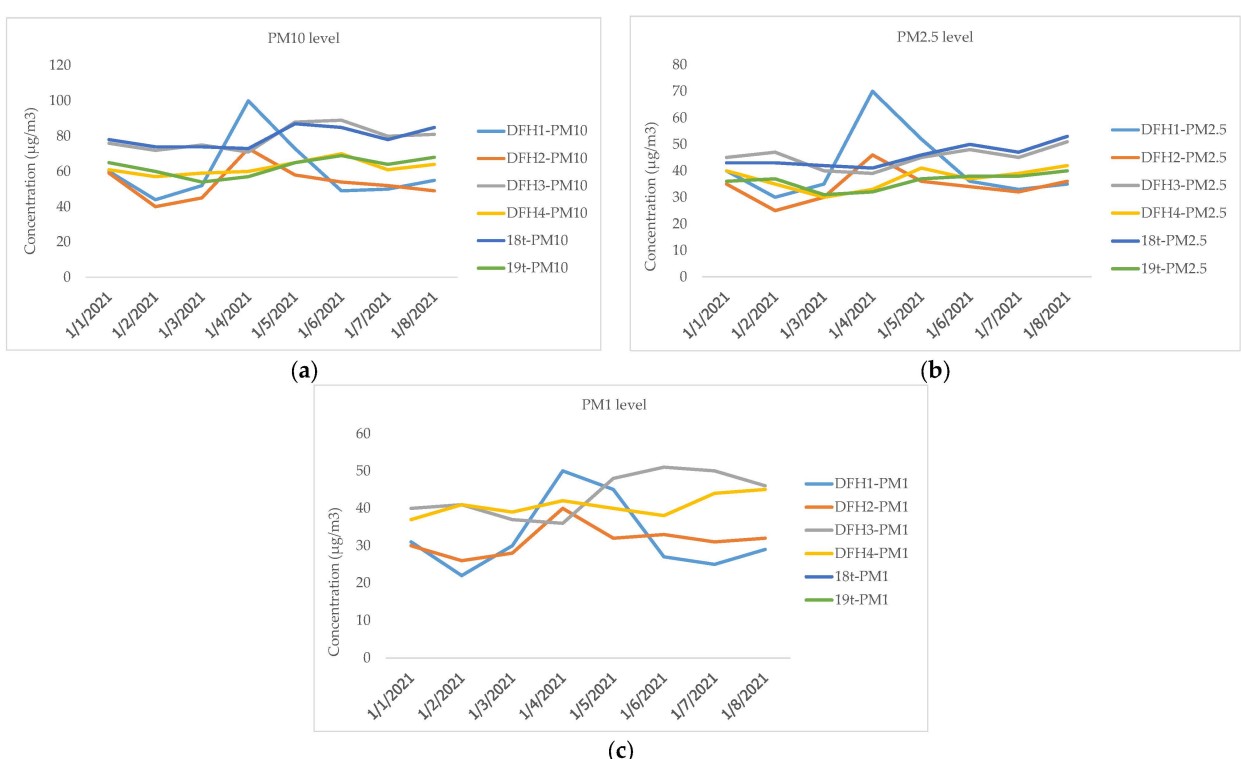

**Figure 3.** The comparison of PM concentrations from LCSs and TEOM: (**a**) $PM_{10}$, (**b**) $PM_{2.5}$, and (**c**) $PM_{1.0}$ ($\mu g/m^3$).

The concentration data from our LCSs (Figure 2) showed a positive correlation with the concentration data from the TEOM (Figures 4 and 5). As shown in Figure 4, the $PM_{10}$ of the LCSs and TEOM are correlated for each site, and the $PM_{10}$ of the LCSs and TEOM scatter plots shows a good correlation for DFH3 ($R^2$ = 0.9(18T), 0.6 (19T)), showing that the DFH3 LCS measures similarly to reference devices. The LCS DFH4 and TEOM scatter plots have a positive $R^2$ correlation of 0.7 (18T) and 0.6 (19T). However, the data from the LCSs at the DFH1 and DFH2 sites were not correlated with that from the PCD station, likely due to the large distance between the locations of these LCSs and the PCD station. The relationship between $PM_{2.5}$ from the LCSs and TEOM was also plotted. Figure 5 shows a low correlation between $PM_{2.5}$ from LCSs DFH1 and DFH2 and TEOM. The above has already been explained. $R^2$ = 0.2, 0.0 (18T), 0.2, and 0.1 (19T), respectively. However, the correlation between DFH3 and DFH4 shows that the LCS DFH3 measures similarly to reference devices ($R^2$ = 0.7 (18T), 0.9 (19T)). Finally, the LCS DFH4 and TEOM scatter plots display positive $R^2$ correlations of 0.44 (18T) and 0.7 (19T), respectively. DFH3 and DFH4 had a positive correlation with TEOM because they are close to the reference monitor locations. Larger distances between the LCS and TEOM may lead to larger differences in simultaneous readings. In addition to site factors, field testing may reveal distinct correlations between emission sources (residential and transportation). In terms of PMS7003 correlation with reference stations, Bulot et al. [62] found an $R^2$ ($R^2$ > 0.6) that is extremely close to our investigation (DFH3, DFH4). LCS testing was conducted during 1–8 December 2021. The average concentrations of $PM_{10}$, $PM_{2.5}$, and $PM_{1.0}$ at the four monitoring sites were 53–79 µg/m$^3$, 34–45 µg/m$^3$, 40–51 µg/m$^3$, respectively. The PCD station was unable to collect both particle sizes ($PM_{10}$ and $PM_{2.5}$) because of an instrument limitation; thus, only the daily averages of the $PM_{10}$ and $PM_{2.5}$ concentrations measured at DFH1 to DFH4 and PCD were compared. The average concentrations of $PM_{10}$ and $PM_{2.5}$ from the 18T station were 79 µg/m$^3$ and 46 µg/m$^3$, respectively. The 19T station gave average concentrations of $PM_{10}$ and $PM_{2.5}$ of 63 µg/m$^3$ and 36 µg/m$^3$, respectively.

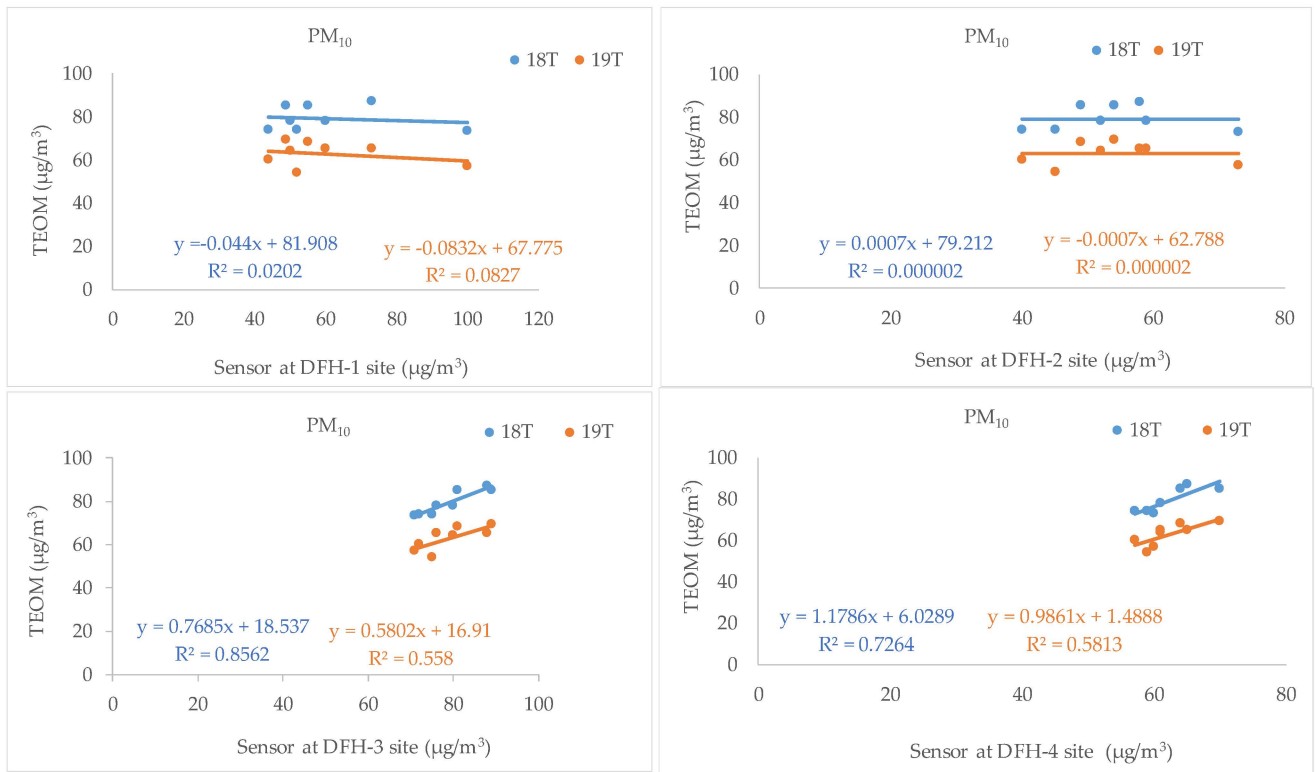

**Figure 4.** The correlation between LCSs and TEOM values for $PM_{10}$.

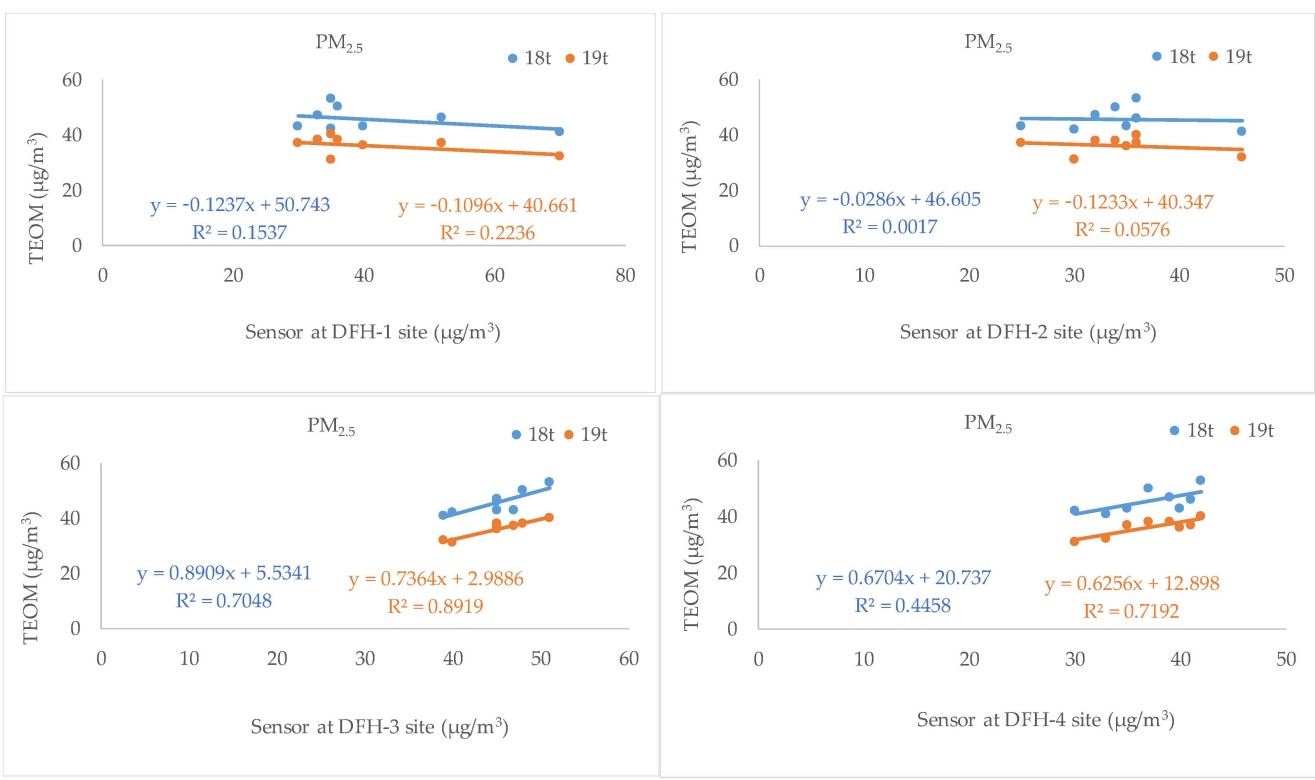

**Figure 5.** The correlation between LCSs and TEOM values for $PM_{2.5}$.

### 3.2. Daily Results

The daily average $PM_{10}$, $PM_{2.5}$, and $PM_{1.0}$ concentrations at the four monitoring sites were measured from 9 December 2021 to 30 April 2022. The results from the coarse and fine-particulate matter concentrations provided values lower than those from TEOM. SPSS analysis was used to determine the difference in the concentration between the LCSs and TEOM. The statistical results are illustrated in Table 2; if the *t*-stat value was less than the *t* critical value, then the hypothesis was acceptable. The mean concentration between the LCSs and TEOM is not statistically significantly different because the value at the significant difference level is more than 0.05 (except DFH-2 vs 18T for $PM_{10}$ and DFH-4 vs 18T for $PM_{2.5}$). The negative number indicates that the average concentration of TEOM is lower than that of the LCSs. The correlation between the LCSs and TEOM values are shown in Figures 4 and 5. The following figures (Figures 6–10) show the time-series trend of the daily average from the important PM contribution at the monitoring sites. The distributions of the $PM_{10}$, $PM_{2.5}$, and $PM_{1.0}$ concentrations at the monitoring site in Samut-prakarn, Thailand, are shown. Each bar represents the average values from the monitoring sites (Figures 6–10). From these results, the $PM_{10}$, $PM_{2.5}$, and $PM_{1.0}$ concentration data from the LCSs during December 9 to 31, 2021, at DFH1 and DFH3, had highest concentration, as shown in Figure 6a,c. The maximum concentrations of $PM_{10}$ reached 129 µg/m³ (22–23 December 2021, green color). The $PM_{2.5}$ concentrations at the DFH1 and DFH3 were 100 and 96 µg/m³ (orange color), respectively, followed by the $PM_{1.0}$ concentrations of 75 and 72 µg/m³ for DFH1 and DFH3 (22 December 2021, as presented in the blue color), respectively. In January 2022, the daily average concentrations of $PM_{10}$ were the highest at DFH3 (63 µg/m³) and DFH4 (58 µg/m³) as shown in Figure 7c,d, and the highest $PM_{2.5}$ concentrations were found at DFH1 (42 µg/m³) and DFH3 (41 µg/m³). Additionally, the $PM_{1.0}$ concentrations had a similar trend at DFH1 (33 µg/m³) and DFH3 (35 µg/m³). Overall, in February, the detection from the LCSs was lowest at DFH4 (Figure 8d) and highest at DFH1 (Figure 8a). The graph shows a decreasing concentration of the data in March. The lowest concentration for all particle sizes from the LCSs was found at DFH2 (Figure 9b) and DFH3 (Figure 9c). The highest concentration for all particle sizes was found

at DFH4 (Figure 9d). Based on the results from April 2022, the $PM_{10}$ concentration was maximum at DFH1 (67 $\mu g/m^3$), as shown in Figure 10a, followed by the $PM_{2.5}$ concentration maximum at 48 $\mu g/m^3$ (Figure 10a), and the highest concentration of $PM_{1.0}$ was 33 $\mu g/m^3$ (Figure 10a).

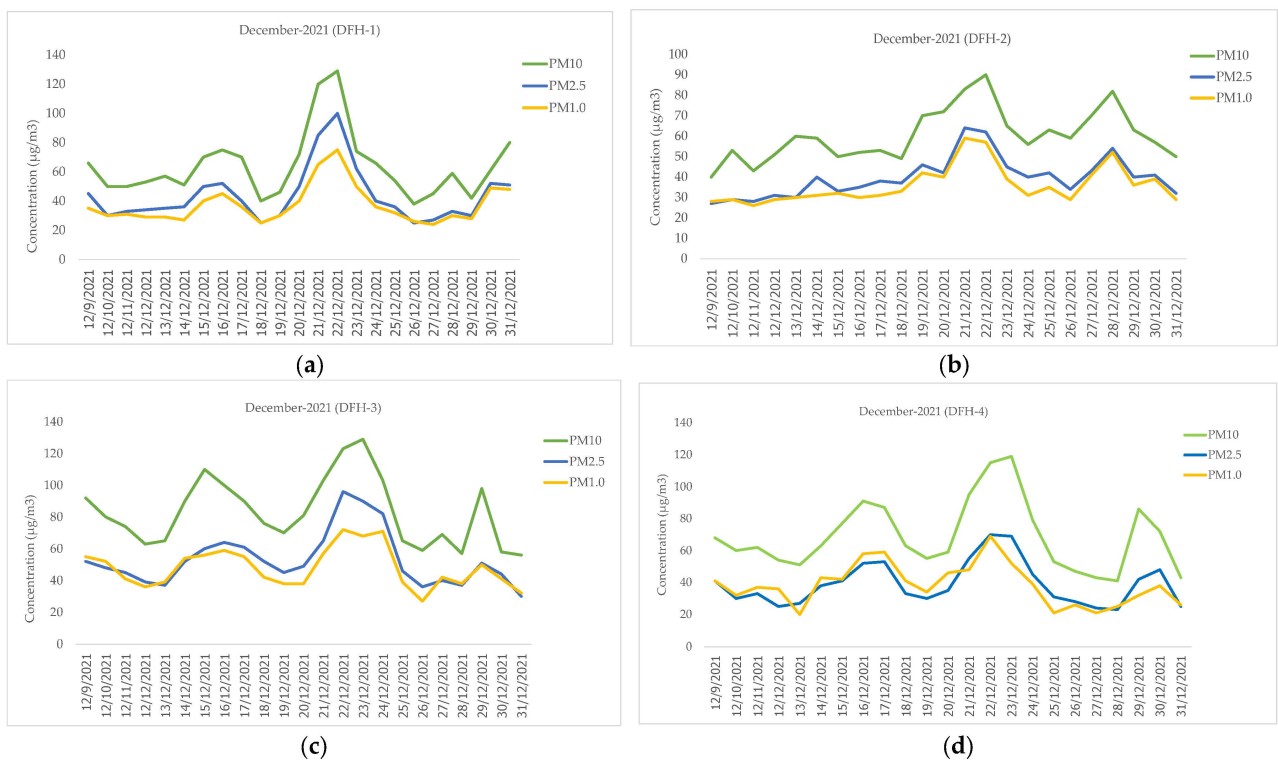

**Figure 6.** $PM_{10}$, $PM_{2.5}$, and $PM_{1.0}$ ($\mu g/m^3$) measured from LCSs during the period from 9 to 31 December 2021: (**a**) DFH-1, (**b**) DFH-2, (**c**) DFH-3, and (**d**) DFH-4.

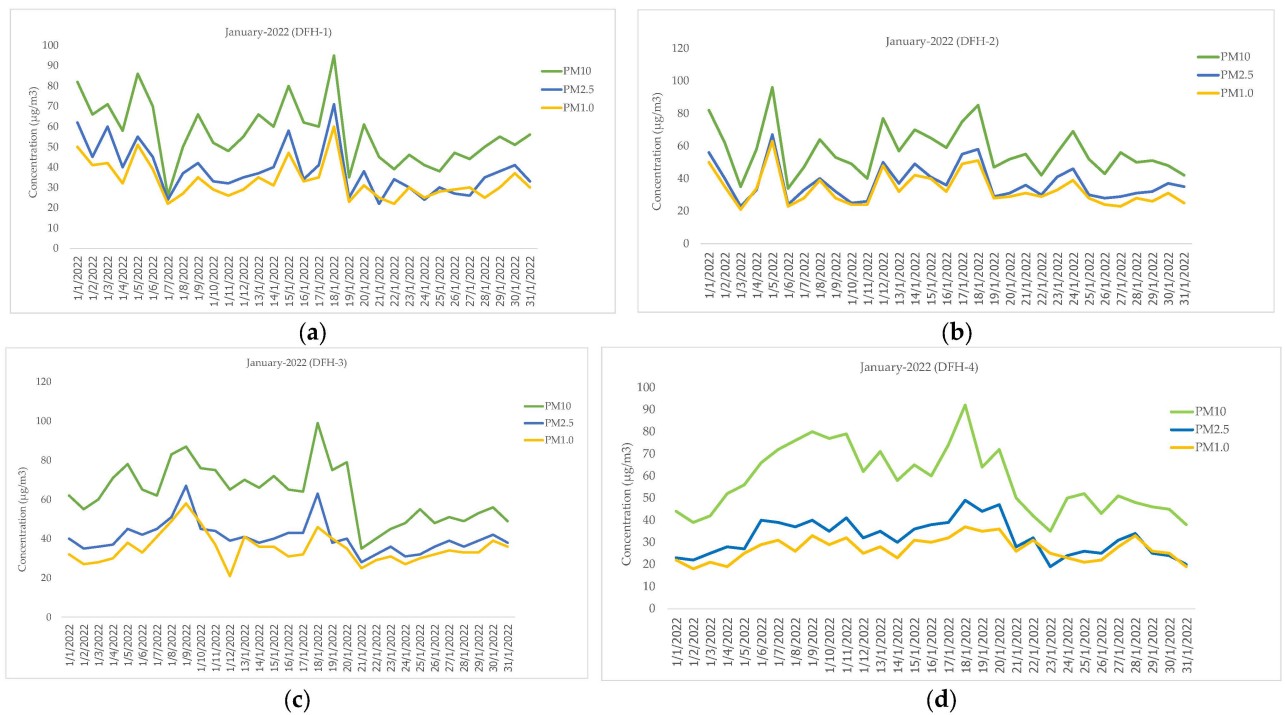

**Figure 7.** $PM_{10}$, $PM_{2.5}$, and $PM_{1.0}$ ($\mu g/m^3$) measured from LCSs during the period from 1 to 31 January 2022: (**a**) DFH-1, (**b**) DFH-2, (**c**) DFH-3, and (**d**) DFH-4.

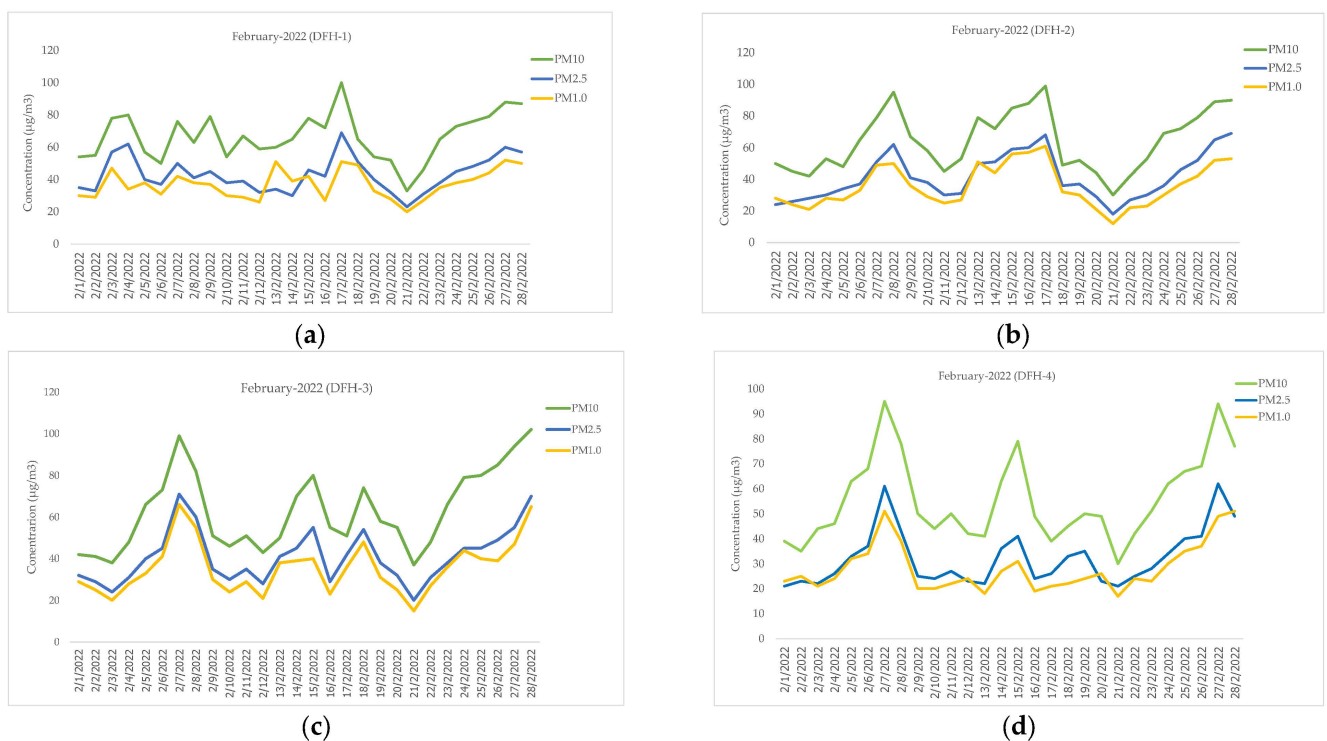

**Figure 8.** PM$_{10}$, PM$_{2.5}$, and PM$_{1.0}$ (μg/m$^3$) measured from LCSs during the period from 1 to 28 February 2022: (**a**) DFH-1, (**b**) DFH-2, (**c**) DFH-3, and (**d**) DFH-4.

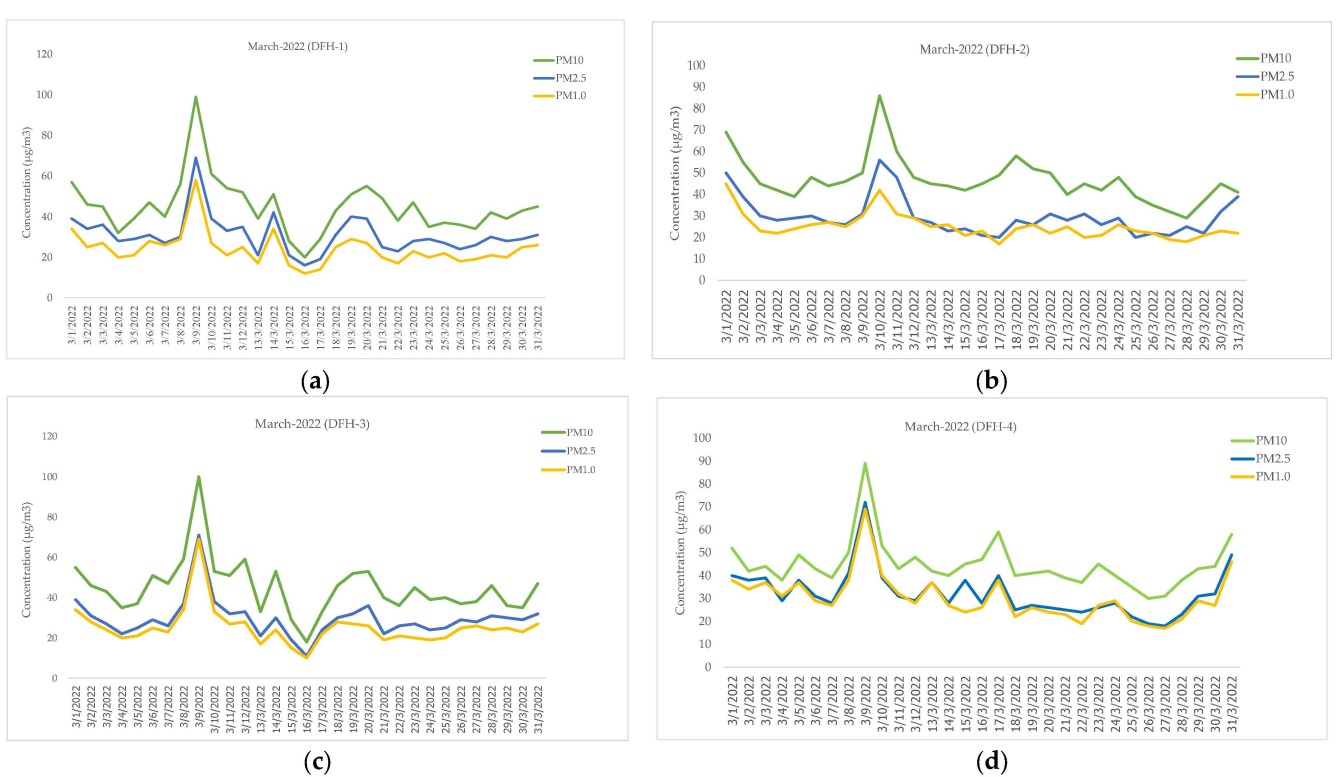

**Figure 9.** PM$_{10}$, PM$_{2.5}$, and PM$_{1.0}$ (μg/m$^3$) measured from LCSs during the period from 1 to 31 March 2022: (**a**) DFH-1, (**b**) DFH-2, (**c**) DFH-3, and (**d**) DFH-4.

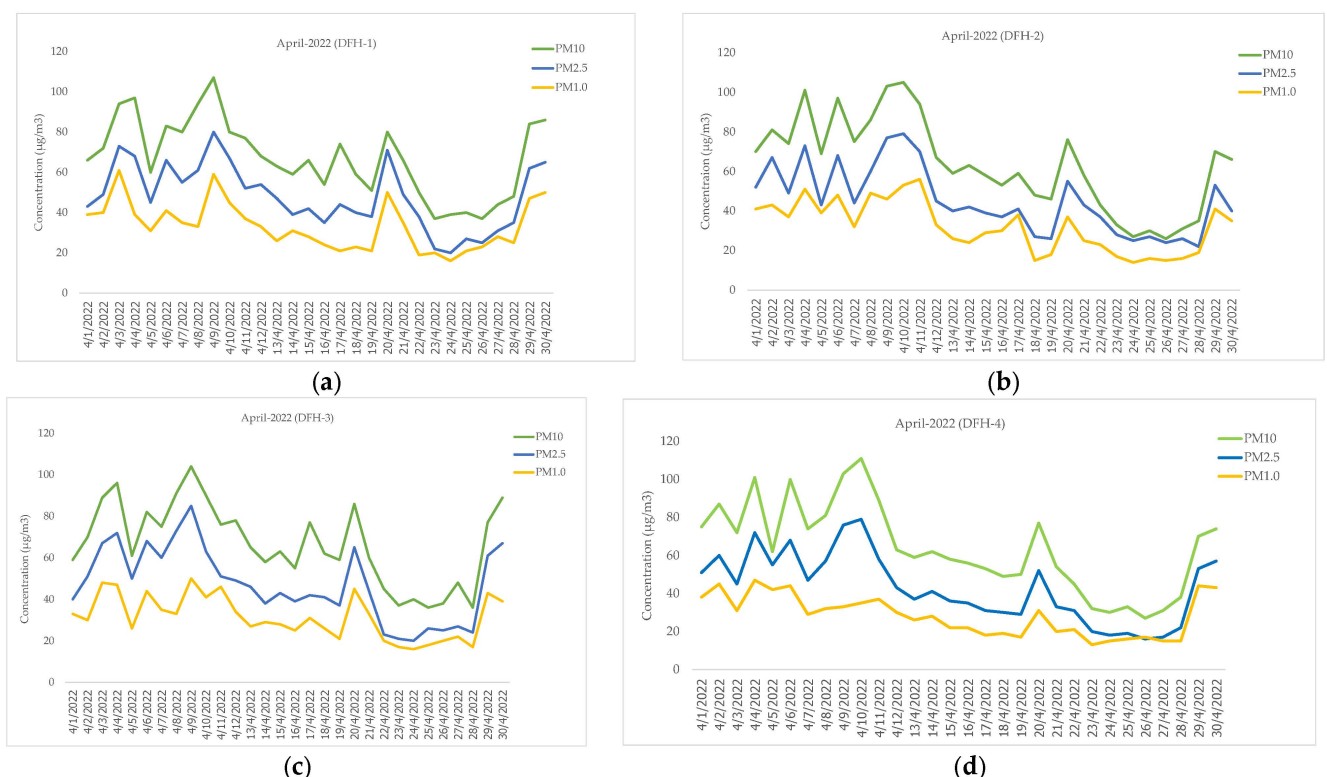

**Figure 10.** PM$_{10}$, PM$_{2.5}$, and PM$_{1.0}$ (µg/m$^3$) measured from LCSs during the period from 1 to 30 April 2022: (**a**) DFH-1, (**b**) DFH-2, (**c**) DFH-3, and (**d**) DFH-4.

*3.3. Highlighted Mass Concentration of Size Distribution*

The particle size distributions in the Samutprakarn province are presented in Figures 6–10. The highest mass concentration was PM$_{10}$, in all monitoring sites, followed by PM$_{2.5}$ and PM$_{1.0}$, respectively. LCSs have been widely used for fine particle analysis in many regions in South-East Asia over the past decade; LCSs provide low-cost, small size capability, easily used instrumentation, and PM mass data. The daily average concentrations of PM$_{2.5}$ at the DFH-1 were 48 ± 16 µg/m$^3$ (April 2022) and 44 ± 19 µg/m$^3$ (December 2021) as shown in Table 3. The daily average concentrations PM$_{2.5}$ at DFH-2 were 45 ± 17 µg/m$^3$ (April 2022) and 42 ± 15 µg/m$^3$ (February 2022). The corresponding values at the DFH-3 were 53 ± 17 µg/m$^3$ (December 2022) and 47 ± 15 µg/m$^3$ (April 2022). The average daily concentrations of PM$_{2.5}$ at the DFH-4 were 43 ± 18 µg/m$^3$ (April 2022) and 39 ± 14 µg/m$^3$ (December 2021), respectively. The average concentration of PM$_{2.5}$ in Thailand was attributed to a high level in the dry season. Moreover, Chunitiphisan et al. [44] revealed that the ambient PM$_{2.5}$ in Northern Thailand had considerably increased, and the mass level in the wet season was found to be lower than that in the dry season at the sampling place [33,64].

In this work, the study period was designed to be during the haze season in Thailand to estimate the air quality data and sensor performance in different urban environments, and the LCS were placed at four locations in Samutprakarn. The differences in the PM$_{10}$, PM$_{2.5}$, and PM$_{1.0}$ concentrations from LCS potentially depended on the location performance. The results from the fine particle in this study showed a high concentration in December at a residential area, and its value was higher than some reports in Table 3. Low levels of PM$_{2.5}$ were found in the DustBoy sensor when compared with the results from this study [45]. However, research reports from Myanmar and Vietnam showed higher results than our study. The PM$_{2.5}$ concentrations in Myanmar in Yangon city were higher in morning (91 ± 37 µg/m$^3$) and lower in evening (60 ± 22 µg/m$^3$) [2]; moreover, the PM$_{2.5}$ concentrations in Mandalay city were higher in the summer (94 ± 10 µg/m$^3$) than in winter (53 ± 2 µg/m$^3$) [48]. The sensors from Myanmar were calibrated before testing (Pocket PM$_{2.5}$ Sensor and AS-LUNG-O). The PM$_{2.5}$ concentrations reported from Vietnam were

the average values. At the first site, Panasonic $PM_{2.5}$ sensors were used in Hanoi and Thai Nguyen Province, and the $PM_{2.5}$ concentrations were measured hourly for three sites and were determined to be 58, 55, and 54 $\mu g/m^3$ [51]. At the second location, low levels of $PM_{2.5}$ were observed in Ho Chi Minh City, Vietnam (Plantower PMS 3003); the average concentration data of $PM_{2.5}$ obtained from the two sensors were 34 $\mu g/m^3$ for sensor 1 and 34 $\mu g/m^3$ for sensor 2 [65].

**Table 3.** Level of $PM_{2.5}$ from LCS at different locations in various study.

| Studied Location, Country | Characteristic | $PM_{2.5}$ Level ($\mu g/m^3$) | Sensor Device | Reference |
|---|---|---|---|---|
| Samutprakarn, Thailand | Residential<br>Semi-residential<br>Commercial<br>Residential | $48 \pm 16$<br>$45 \pm 17$<br>$53 \pm 17$<br>$43 \pm 18$ | PMS7003 | This study |
| Bangkok, capital of Thailand | Urban | $36 \pm 12$ | DustBoy | [45] |
| Nan, Northern Thailand | Urban | <5–37<br>(flight track) | Plantower PMS 3003<br>(on Drone) | [44] |
| Bangkok, capital of Thailand<br>Chiangmai, Northern Thailand<br>Ubon Ratchathani, Northeast Thailand | Urban | $27 \pm 18$ (dry season)<br>$14 \pm 11$ (wet season)<br>$41 \pm 29$ (dry season)<br>$11 \pm 8$ (wet season)<br>$39 \pm 27$ (dry season)<br>$18 \pm 16$ (wet season) | PMS7003, Plantower | [64] |
| Yangon, Myanmar | Urban | Kamayut,<br>Morning $91 \pm 37$<br>Evening $60 \pm 22$ | Pocket $PM_{2.5}$ Sensor | [2] |
| Mandalay, Myanmar | Urban | Summer $94 \pm 10$<br>Winter $53 \pm 2$ | AS-LUNG-O | [57] |
| Taipei, Taiwan | Urban | Location A $17 \pm 9$<br>Location B $11 \pm 4$ | AS-LUNG-O | [7] |
| Taipei, Taiwan | Urban (households) | $18 \pm 11$ | AS-LUNG-O | [66–68] |
| Vietnam Hanoi and Thai Nguyen Province | Urban | Hourly: three sites, 58, 55, and 54 | Panasonic $PM_{2.5}$ sensors | [51] |
| Ho Chi Minh City, Vietnam | Urban | Sensor 1: 34<br>Sensor 2: 34 | Plantower PMS 3003 | [65] |
| Jakarta, Indonesia | Urban | 50–65 | Edimax AirBox<br>AI-1001W V3 | [53] |
| Košice, Slovak republic | Urban and rural | 20–80 | SPS30<br>SEN54 | [3] |
| U.S.–Mexico border | Urban<br>Residential<br>Residential<br>household | 9<br>7<br>7 | BlueSky™ Air Quality<br>Monitors, TSI Inc.<br>Shoreview, MN, USA<br>(Model:8143) | [6] |

*3.4. Temperature and RH Sensor*

In this study, the air temperature and RH were monitored using the LCSs. Air temperature performance of LCSs is shown in Figure 11; good agreement was observed between our sensor and TMD. The RH from the LCSs was consistently lower than the results from TMD. The LCS testing occurred during December 2021 (1–8 December 2021). Figure 11a shows the air temperature data. The trend of air temperature from all LCSs was similar to the recoded values from the TMD. The results from the temperature sensor at the DFH-1 (26 °C) site were similar to the DFH-2 site (27 °C). The average air temperature at the DFH-3 and the DFH-4 site showed a general temperature of ~26 °C. Additionally, the RH from our

sensor (DFH-1 = 59%; DFH-2 = 60%; DFH-3 = 64%; DFH-4 = 63%) showed lower values than TMD (84%) for all sites. Our sensor devices for air quality data contained temperature and RH sensors. The DHT22 sensor is used for temperature and RH measurement [44]. Figures 12–16 display the study results of temperature and RH over the studied time intervals using LCS. The average air temperature of the LCS ranged between 26 °C and 31 °C, with a RH of 65 to 78%. Bulot et al. [62] observed a strong correlation between PMS7003 and PMS5003 and the background station when the RH was similar (76–98% RH) to the reference device. Moreover, the air temperature and RH can influence PM concentrations. Ly et al. [51] reported that the $PM_{2.5}$ level was associated with meteorological factors, such as wind speed, wind direction, air temperature, RH, and atmospheric pressure. Air temperature had a strong effect on $PM_{2.5}$ in the winter and had a weak effect on $PM_{2.5}$ in summer [33]. Dejchanchaiwong et al. [64] found that low PM concentrations and high RH levels had a substantial impact on LCS performance. Specifically, during the wet season, LCS showed higher relative inaccuracy than during the dry season. In our results, PM mass concentrations were raised at low air temperatures utilizing LCS from December 2021 to February 2022, as shown in Figures 12 and 13. It was found that weather parameters in this period could cause sufficient variations in temperature, like temperature inversion, to reduce the height of the atmospheric boundary layer and dust diffusion, therefore increasing dust concentration. There is also extensive biomass burning from agricultural burning in preparation for the rice planting season in the winter, along with the long-range transport of air pollutants from neighboring countries such as Laos, Vietnam, and Myanmar [61,62]. In comparison, the lowest RH levels were reported in December 2021 and January 2022.

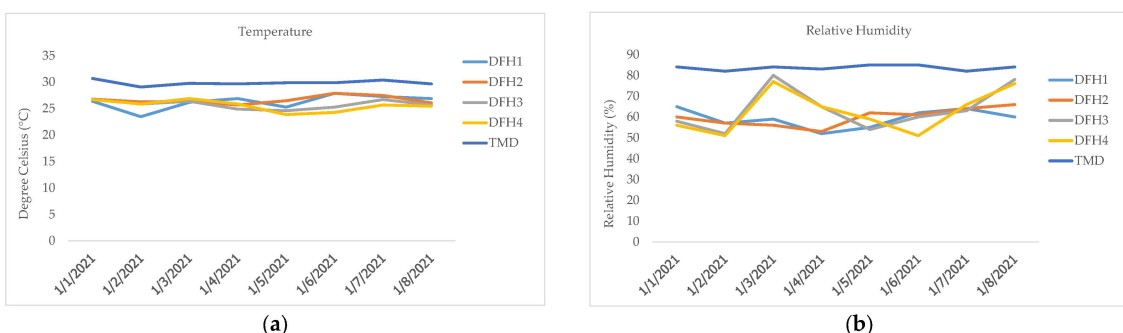

**Figure 11.** The comparison of weather from LCSs and TMD: (**a**) air temperature and (**b**) RH.

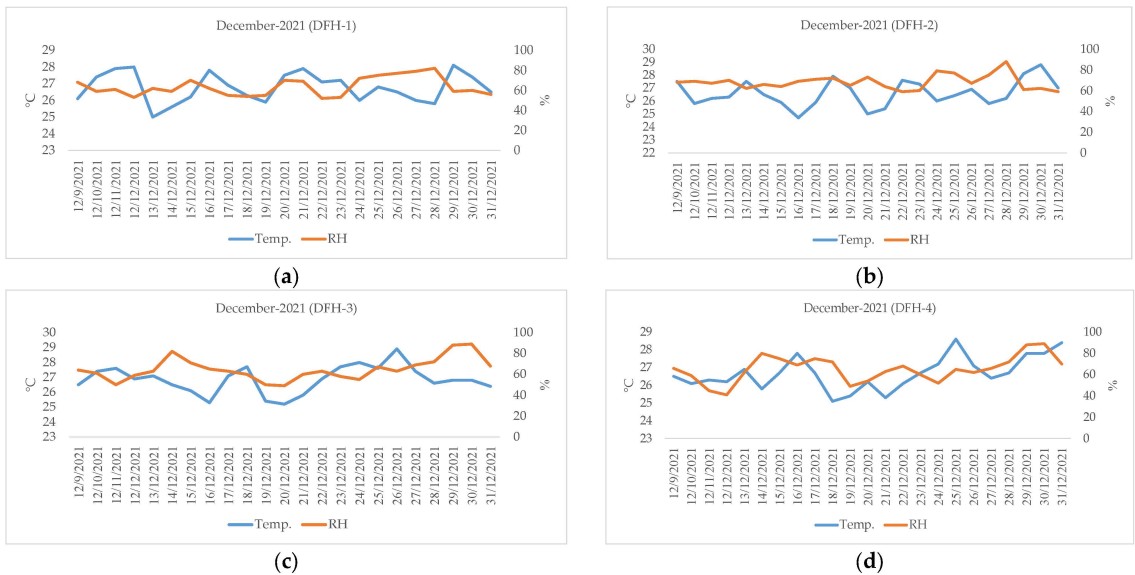

**Figure 12.** Air temperature (°C) and RH (%) measured from LCSs during the period from 9 to 31 December 2021: (**a**) DFH-1, (**b**) DFH-2, (**c**) DFH-3, and (**d**) DFH-4.

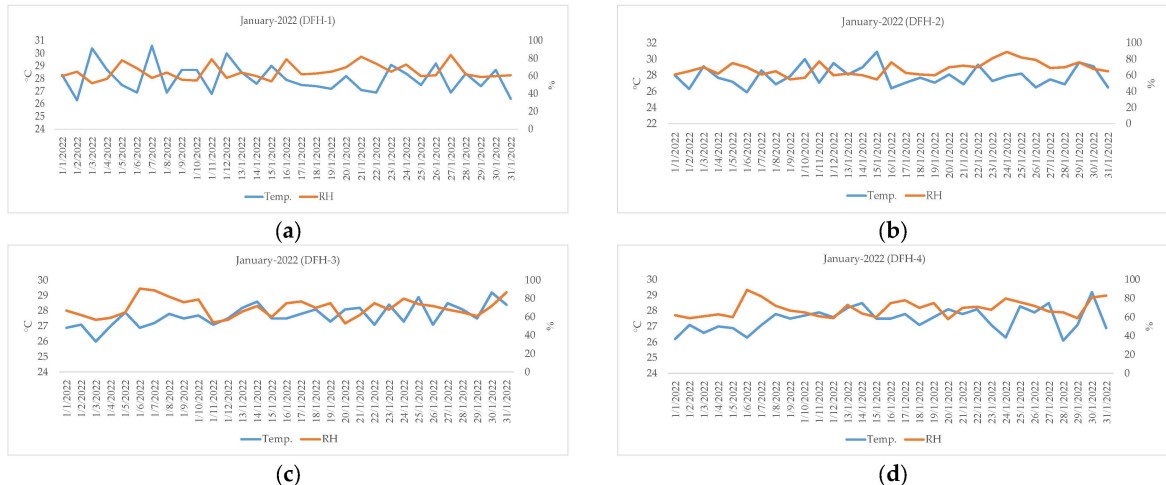

**Figure 13.** Air temperature (°C) and RH (%) measured from LCSs during the period from 1 to 31 January 2022: (**a**) DFH-1, (**b**) DFH-2, (**c**) DFH-3, and (**d**) DFH-4.

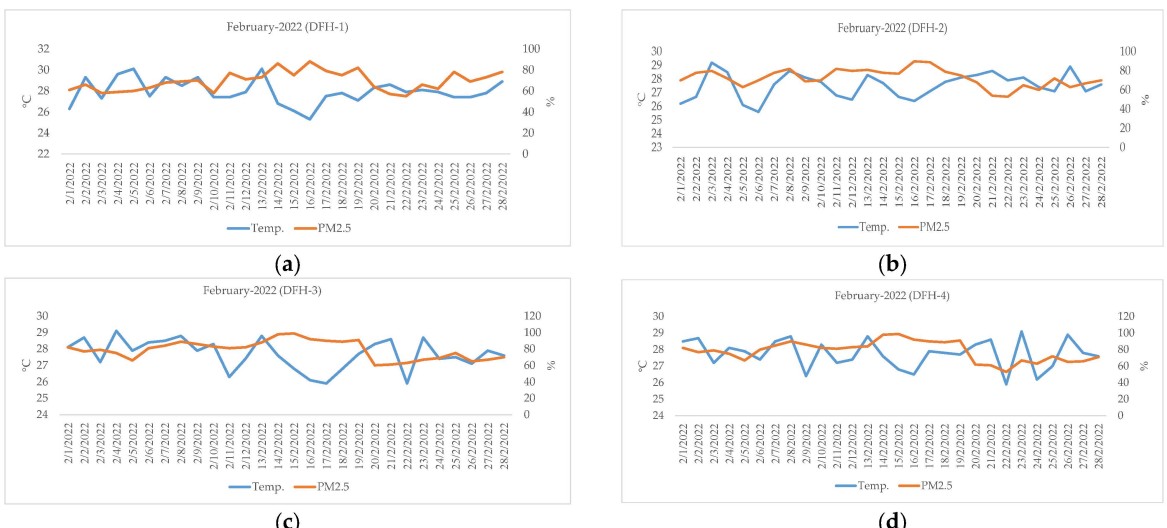

**Figure 14.** Air temperature (°C) and RH (%) measured from LCSs during the period from 1 to 28 February 2022: (**a**) DFH-1, (**b**) DFH-2, (**c**) DFH-3, and (**d**) DFH-4.

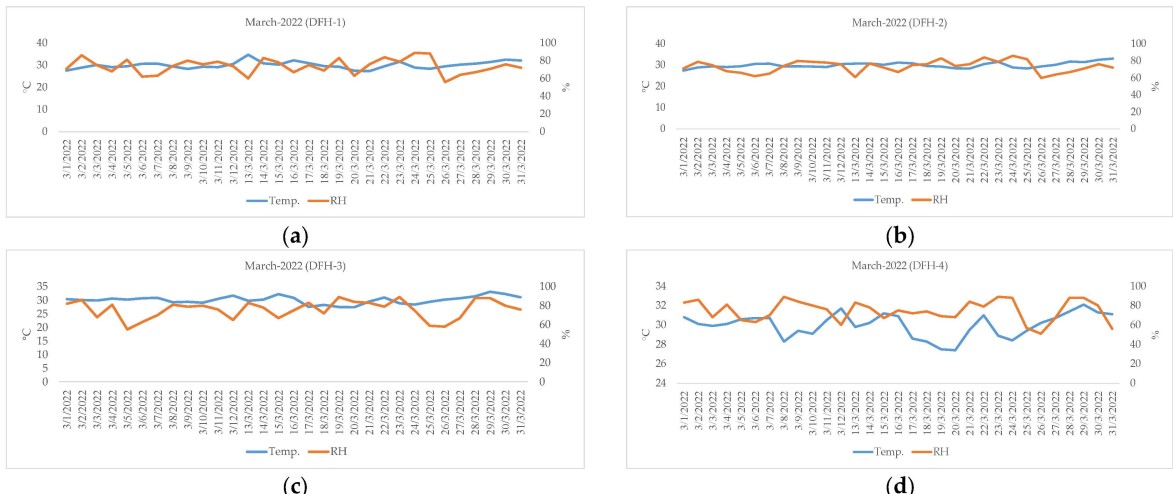

**Figure 15.** Air temperature (°C) and RH (%) measured from LCSs during the period from 1 to 31 March 2022: (**a**) DFH-1, (**b**) DFH-2, (**c**) DFH-3, and (**d**) DFH-4.

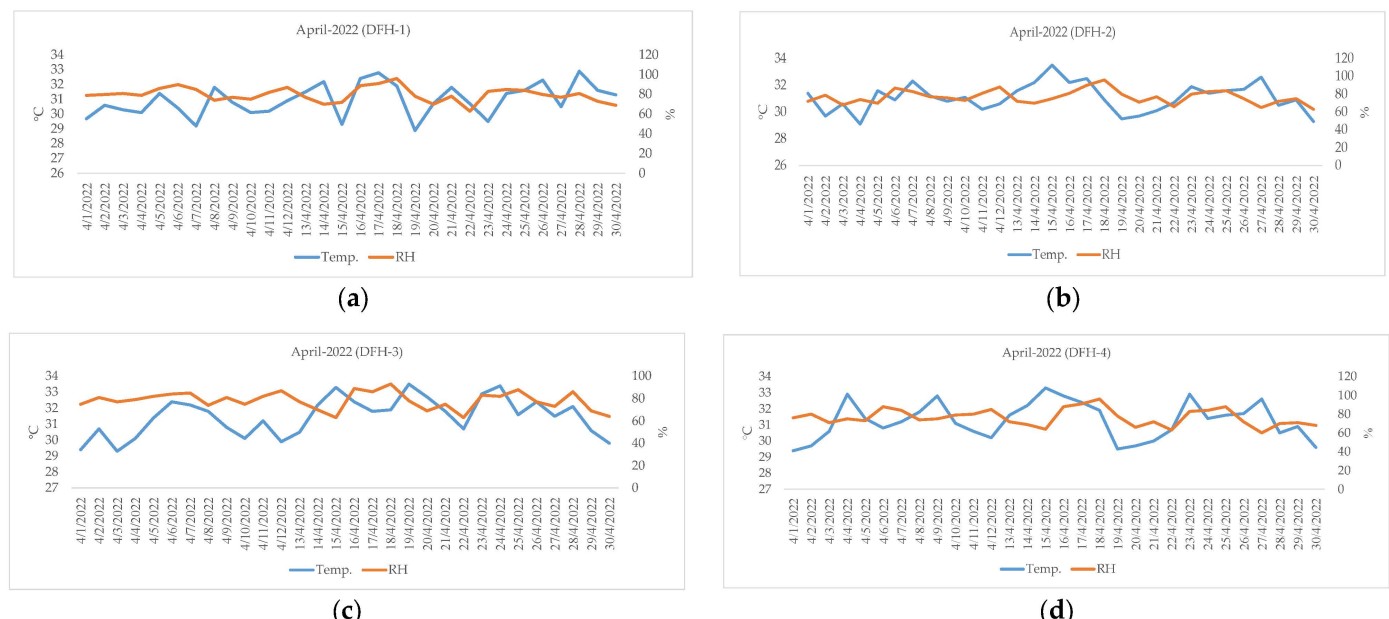

**Figure 16.** Air temperature (°C) and RH (%) measured from LCSs during the period from 1 to 30 April 2022: (**a**) DFH-1, (**b**) DFH-2, (**c**) DFH-3, and (**d**) DFH-4.

### 3.5. Air Quality Index (AQI)

The air quality index (AQI) data used to report pollution concentration in Thailand was calculated from the LCSs in all sites; their values and color code of each month are shown in Figure 17. Daily $PM_{10}$ concentrations (calculated from an average over 24-h) at all LCSs were selected to calculate the AQI equation because $PM_{10}$ had higher values than the other particle sizes. Based on our results, $PM_{10}$ played a significant role in the AQI (Figure 17a–e). Since the $PM_{2.5}$ and $PM_{1.0}$ levels were lower than the $PM_{10}$ level, our results could not indicate their hazard level. Different pollutants have different toxicities and health risks; consequently, the AQI is predicted from estimates of the pollutants $CO$, $NO_2$, $O_3$, $SO_2$, $PM_{10}$, and $PM_{2.5}$ and is calculated from the average concentrations of the six pollutants. Notably, the fine particles are thought to have a greater influence on the human body because they can penetrate deeper into the human respiratory system. The complete data were imported to GIS and then created in map form. The PM mass concentration was higher in December to February than March–April. As a result, the PM mass concentrations for AQI were estimated using the Thailand AQI standard equation. A pattern of geographic contribution and large variation was shown for the various monitoring sites. The AQI values of Samutprakarn province were visualized via PM mapping. This research revealed that the months of December through February were yellow and green, respectively, while March and April were blue and green at our monitoring sites. The AQI values showed good to moderate results. During our study period, the results demonstrated satisfactory air quality from January to April 2022, as indicated by blue (Figure 17d) and green (Figure 17e), respectively. However, the DFH3 showed higher levels of pollutants, and the pollutants were likely from an anthropogenic contribution since this site was a residential area, as shown in Figure 17a).

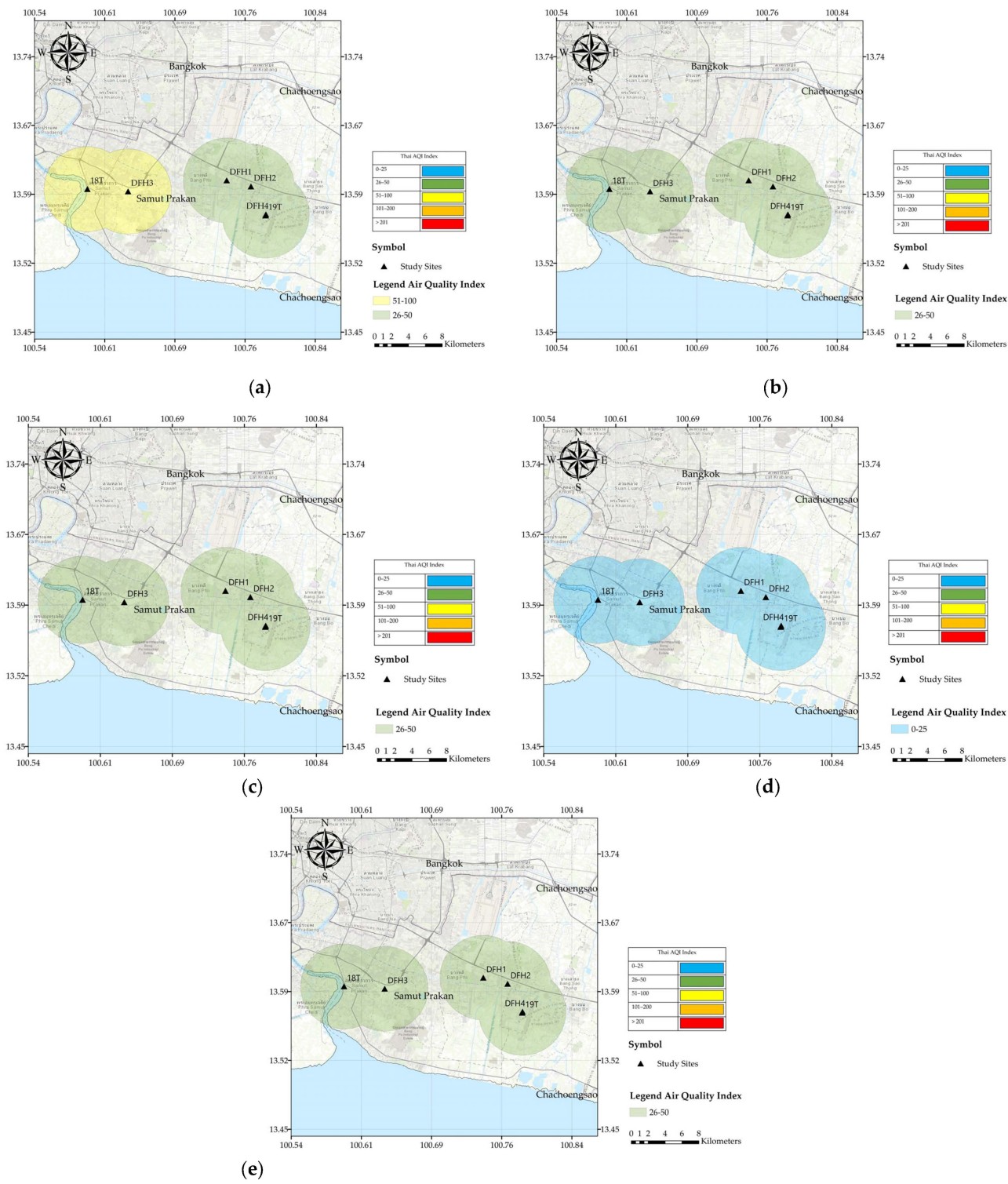

**Figure 17.** AQI mapping: (**a**) December 2021; (**b**) January 2022; (**c**) February 2022; (**d**) March 2022; (**e**) April 2022.

## 4. Conclusions

The concept of LCS for monitoring particles in urban ambient air that includes the size range $PM_{10}$, $PM_{2.5}$, and $PM_{1.0}$ was presented in this paper. The LCSs at all four (DFH1, DFH2, DFH3, DFH4) monitoring sites in urban areas detected $PM_{10}$, $PM_{2.5}$, and $PM_{1.0}$ during the monitoring periods. Coarse ($PM_{10}$) and fine particles ($PM_{2.5}$, $PM_{1.0}$) were collected from the LCSs and compared to a TEOM monitor; a TEOM is a reference

instrument approved by the Pollution Control Department, Thailand. We have provided evidence that the LCSs (PMS7003) can detect $PM_{10}$, $PM_{2.5}$, and also $PM_{1.0}$. Most PM concentrations from the LCSs were lower than those from the TEOM. The LCSs were employed continually in an urban area of Samutprakarn Province. Data were collected over a monitoring period of five months (of course a full year would be ideal). Furthermore, temperature and RH sensor were also measured using the DHT22 sensor. Temperature and RH trends from all LCSs showed less sensitivity and accuracy than the TMD.

The observation results revealed a high level of PMs at DFH3, a residential site, while other locations had a lower level of PMs. These results potentially indicated that the level of PMs was affected by the site features. We understand that our study has limitations. Only a few minor sites were chosen for ambient air sampling and monitoring using LCSs. Furthermore, weather conditions had a substantial impact on PM levels. The observation results of PM mass concentration were higher in December to February than March to April. As a result, the PM mass concentrations for AQI were estimated using the Thailand AQI standard equation. A pattern of geographic contribution and large variation was shown for the various monitoring sites. The AQI values of Samutprakarn province were visualized via PM mapping. This research revealed that the months of December through February were yellow and green, respectively, while March and April were blue and green at our monitoring sites. The AQI values revealed good to moderate findings, indicating that there is no health impact at the study sites. Finally, our findings proved the ability of our LCSs to quantify ambient particles and their spatial distribution.

**Supplementary Materials:** The following supporting information can be downloaded at: https://www.mdpi.com/article/10.3390/atmos15030336/s1, Figure S1: TEOM at air monitoring station of Pollution Control Department (PCD) (18T); Figure S2: TEOM at air monitoring station of Pollution Control Department (PCD) (19T); Table S1: List of acronyms used in study; Table S2: Specific of Tapered Element Oscillating Microbalance (TEOM) ambient particulate monitor.

**Author Contributions:** S.R., conceptualization, data collection, investigation, methodology, formal analysis, visualization, writing—original draft, writing—review and editing; S.C., conceptualization, data collection, methodology, visualization; S.J., investigation, visualization, writing—original draft, writing—review and editing; P.T., data curation, formal analysis, writing—review and editing; W.P., writing—review and editing; T.C., writing—review and editing; Y.B., conceptualization, data collection, investigation, methodology, formal analysis, visualization, writing—original draft, writing—review and editing. All authors have read and agreed to the published version of the manuscript.

**Funding:** This research received no external funding.

**Institutional Review Board Statement:** Not applicable.

**Informed Consent Statement:** Not applicable.

**Data Availability Statement:** The data presented in this study are available in file of Supplementary Materials.

**Acknowledgments:** The authors would like to thank the house owner for the use of their property for research purposes. Furthermore, this study was supported equipment by the Sustainable Development Technology, Faculty of Science and Technology, Thammasat University.

**Conflicts of Interest:** The authors declare no conflicts of interest.

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
