# Peer review of "Using a Low-Cost Sensor to Estimate Fine Particulate Matter: A Case Study in Samutprakarn, Thailand"

_atmosphere, doi:10.3390/atmos15030336_

Round 1
Reviewer 1 Report
Comments and Suggestions for Authors
Report on manuscript atmosphere-2851500
“Using a Low Cost Sensor to Estimate Fine Particulate Matter: A case Study in Samutprakarn, Thailand”
This manuscript aims to develop a low-cost sensor device for measuring the fine particulate matter (PM2.5), along with some environmental data including temperature and humidity. The manuscript is promising and has the potential to be published but after major revision. I would like the authors to address the following issues:
Comments:
- Title: low cost à Low-cost, A case --> A Case
- Introduction is written in one paragraph. Authors need to revise to organize it in multiple paragraphs to make it more readable.
- In L116, authors started to talk about their motivation for the study, however, the motivation is not very clear. Please extend your motivation and the importance of the study, and how your work adds to the existing knowledge. It would be good to mention some similar studies.
- Figures 4 and 5: too much space is taken by the values of Y and R2. Try to put these values above or below the data point so that we can zoom in on the figure.
- I couldn’t find the reference for the data.
- Too many abbreviations, it will be useful to add an abbreviation table as an appendix at the end of the manuscript.
- Add punctuation symbols in equation 1 and the definitions of the variables.
- L305: results here are presented also in one paragraph, try to split it to organize the findings and make it clearer.
- L342: the text is not readable in this line.
- L354: elaborate on the sentence “the weather parameters influence on PMs concentrations”, and how this is reflected in your results.
- L393: “During our study period, the results showed satisfactory of air quality in January to April 2022.” You may need to add supporting info for this claim.
- Conclusions: the conclusion is not very clear and doesn’t conclude what has been discussed in the paper. Many short statements are presented and need revision. For instance, saying “The results from the low-cost sensors were acceptable” is not enough, you need to support it with the findings of the paper. Same thing in Line 426.
Comments on the Quality of English Languagefew points are mentioned in the report.
Reviewer 2 Report
Comments and Suggestions for Authors
As a paper using a new instrument to study PM concentrations, the validation results of the instrument are very important, but this manuscript does not focus on validating the reliability of the instrument and simply makes a few correlation plots with PCD data and test data, and therefore shows little academic significance. In addition, there are some problems of inconsistency between the graphical presentation and the exposition.
Specific comments:
[1] Introduction: Please reorganize and rewrite the introduction. State the objectives of the work and provide an adequate background, avoiding a detailed literature survey or a summary of the results.
There is no need to go over the basics about PMs. The introduction talks about the sources of PMs, but sources are never mentioned again in subsequent studies and discussions of particulate matter. Please delete the redundancy.
check the manuscript, to avoid expressions such as " [31] showed the measurement " in P2 line 68.
[2] Materials and Methods
P3 line 130, revise the sentence" The permanent population is 130 1,344,875 million people". population includes people, replace "permanent population" with another term, such as resident population.
[3] P6 line 225 "course" should be "coarse"
[4] the third figure in Fig 4 should be DFH-3, not DFH-1
[5] I don't think Fig6-10 are particle size distributions plots, they are just mass concentrations plots.
[6] P15 line 342-343 "good agreement was observed between our sensor and Thai Meteorological Department (TMD). " where can we see this results?
[7]P18 line 424, "This finding highlighted that the months of December to February were hazy, " How can you get this, form the figure 13, the AQI are all green, except DHF3 in yellow (AQI 51-100) which states moderate according line 380,
What is the AQI value that is considered hazy?
Reviewer 3 Report
Comments and Suggestions for Authors
General comment:
In the paper, the authors evaluate Low-Cost Sensors (LCS), and one could argue that this falls within the journal's scope. However, it would be more justified for a journal specifically dedicated to sensors, considering that the precision of measurements is the primary focus of the study, rather than changes occurring in the atmosphere in the context of parameters measured by the sensors. The presented form of the conducted research does not convince me of the accuracy of the specified devices. Specifically, there is a lack of consideration for the readings of two closely placed LCS sensors. While the authors correctly positioned two sensors close to the indicated reference stations, nothing is known about the differences in the readings of the two LCS devices for approximately the same atmospheric conditions.
Although the authors correctly positioned two sensors close to the indicated reference stations, there is a lack of consideration for the readings of two closely placed LCS sensors. The conducted statistical analyses, first and foremost, are basic and do not introduce any new methodology. Simultaneously, the manner in which they are described and presented raises doubts about what was interpreted by the authors based on them. This needs to be rectified urgently.
The overall structure of the article is somewhat unclear, with parts of the introduction mixed into the results and methods, and vice versa. It is crucial to read the entire work and arrange it in a comprehensible manner. Only halfway through the article does it become clear what the authors are constructing and ultimately evaluating.
The graphics and tables are not self-explanatory. In the context of statistical analyses, the "Material and Methods" section does not explain why and for what purpose these specific methods were used. This needs to be restructured and clearly explained.
The authors do not address the topic of processing LCS data in the context of WMO guidelines at all. The spatial analysis aspect is unacceptable and must be discussed in the context of leading scientific works utilizing a dense network of LCS for spatial analyses. Paper in general lacks novelty, how those fings can be applied to the wider scientific audience?
Specific comment:
Introduction:
Please avoid combined citations like in lines 38, 42 and so far! If the study is relevant mention shortly the relevant part and put the direct reference, if not reduce the number! Use a MAXIMUM of 3 combined references, not 5 or 10.
Please exactly specify what type of measurement of LCS is used and what the sensor's accuracy, also please specify in detail the reference station type of measurement – automatic? Gravimetric?
Line 79 – please check word breaking I assume the text was copied from another document with punctuation and other marks. Please reread the whole text and correct it everywhere
Line 88 is that the same sensor you are using? If yes mention the name at the beginning. Please move the sensors’ accuracy analysis and comparison to the methods paragraph. Almost the whole rest of this paragraph should be moved to materials and methods.
An in-depth exploration of the temporal-spatial dynamics of particulate matter (PMx) is crucial for enriching the reader's comprehension of the subject. Although the authors address essential aspects concerning suspended particles, a meticulous examination of temporal variations, encompassing seasonal trends and global geographical disparities, would significantly contribute to a more thorough understanding of this phenomenon. It is recommended to consult recent publications by Prof. Tomasz Danek, particularly those delving into the impact of meteorological factors and terrain on air pollution concentration in Krakow (Scientific Reports), or utilizing big data for PM pattern analysis (Atmosphere). These studies focus on highly concentrated Low-Cost Sensors (LCS), nearly 100 in number. Please refer to these studies and provide comments regarding your spatial coverage and its potential impact on interpretation. Especially concerning my comment on the last figure.
At the end of the introduction please put a paragraph what is the gap in current research and the novelty of your study. This should be mentioned. If the objective was to develop the LCS sensor – what is new added to the scientific world by this? What is the research here? Anyway, I found that the message of developing a new sensor is not clear from the beginning so try to write it in the way that it is obvious.
Please introduce the acronym for low-cost sensors as LCS and use it in the text.
Material and methods
Figure 1 – The map of Thailand is almost not readable please correct, please add a, b, and c to the parts of Figure 1 and explain in the title
Line 135 combustion of what? Source?
Line 145 – rewrite the sentence is hardly understandable. Where are the 18t and 19t stations? Correct them on the figure is hard to find them.
Line 153 – I’m confused. Is it the sensor that you are developing or the sensor you bought from an external supplier? In the next sentence, you are giving more explanation but it should be rewritten more understandably.
Line 164 – How? Please provide a graphic with the ingestion pipeline. Was it ETL or ELT?
Line 178 – Please provide evidence with graphs and statistics.
Equation 1 is obvious
2.6 and 2.7 please elaborate more on what those were done, what’s the purpose of it, why those metrics, etc.
Results,
Figure 2 shows nothing beside the metal box please see my previous comment.
Line 232 what is compared? Averaged? BE specific as I assume that TEOM and your sensor are performing in different measurement intervals.
All figures in this section must be correct. Each figure should clearly describe what is on it, especially what the averaging window and/or sampling interval
Figure 3 – Statistical analysis on just 8 samples is highly risky and essentially cannot be used to draw conclusions about the effectiveness of the sensors
Correlation? What do you mean specifically? Are you referring to the Pearson correlation coefficient or linear regression? Please be specific.
Line 306 – is it possible to have lower values of PM10 than PM2.5 if PM10 consists of PM2.5 as well? I don’t think so, so what’s the point of it? Please avoid statements that are generally true to your interpretation.
Line 320 – why are you mentioning this objective here not in the beginning?
Line 324 – 338 what is the point of mentioning indications of sensors in different locations and comparing if there is nothing mentioned about sources, meteorological factors, and source analysis?
Line 342 – what was the height of reference sensors and compared LCS sensors? It may be an important factor for temperature and humidity.
Line 357 – this statement is true not only in Thailand but also in other climate types – it’s worth mentioning it – again see Danek et.al.
Line 373 – the medical introduction should be done in the introduction, no need to mention health effects again
Also why you are introducing the AQI in results – please move it to the introduction.
Figure 13 – In my undertaking a spatial analysis for the entire province based on one sensor appears to lack practicality and scientific rigor. This becomes especially apparent when assigning the entire southern province the color green, disregarding the proximity of the sensors to neighboring provinces. Such an approach may not yield meaningful insights and may compromise the accuracy and reliability of the spatial analysis.
Please add X and Y to the maps.
Conclusions should be made in points clearly pointing to what the authors prove in the research. Line 413 “was acceptable” based on what if you are not referring to any guidelines like WMO or others.
Comments on the Quality of English LanguageA native speaker should correct the paper
Reviewer 4 Report
Comments and Suggestions for Authors
1. What is the main question addressed by the research?
The main question of the paper” Using a Low Cost Sensor to Estimate Fine Particulate Matter: A case Study in Samutprakarn, Thailand” is to study the pollution caused by PM10 and PM2.5 in four sites in Samut Prakan, Thailand from December 2021 to 15 April 2022, using low-cost sensors.
2. Do you consider the topic original or relevant in the field? Does it
address a specific gap in the field?
The originality of this article consists in develop a low-cost sensor device for measuring the fine particulate matter (PM2.5), PM10 and environmental data including temperature and humidity. Statistical analysis was done using SPSS 28 and ArcMap version 10.8 software was employed to manipulate map data for the study area.
3. What does it add to the subject area compared with other published
material?
This article does not draw a parallel with other studies conducted in the area of Samutprakarn, Thailand or with other scientific researches in the area of Southeast Asia.
4. What specific improvements should the authors consider regarding the
methodology? What further controls should be considered?
Many other types of tests and verifications and data comparisons are needed, but considering the available budget and the fact that the area is not one that can allocate funds for research, I believe that this study can be the basis for other future research.
5. Are the conclusions consistent with the evidence and arguments presented
and do they address the main question posed?
The conclusions are clear and express the basic ideas resulting from the current study.
6. Are the references appropriate?
The 56 references are eloquent and appropriate for the article
7. Please include any additional comments on the tables and figures.
The 13 figures included in the current study are clear and easy to understand
Have a nice day,
Round 2
Reviewer 1 Report
Comments and Suggestions for Authors
I thank the authors for the revision. Most of my comments have been considered in the revised version. However, while reading the new version, I spotted a couple of old (not considered or justified) or new comments.
- Add a comma after equation one and start the next line with small letters.
- Add TEOM to the supplementary file.
- The authors mentioned that “PMS7003 typically has a strong correlation with the reference instrument and good reproducibility”, please elaborate on the reason for such behavior.
- In L.305: the authors mentioned that The PM concentrations from the low-cost sensors and TEOM are shown in Figure 3, however, these figures are satisfactorily discussed in the text. Authors need to add more explanation about Figure 3, Figure 4, and Figure 5.
- The authors have successfully extended the introduction to provide additional examples and enhance the overall context of their research. However, it is noteworthy that the discussion about solar radiation is missing. Including a brief discourse on solar radiation could further enrich the background of the study. I recommend the authors to consider incorporating a few sentences on this topic by giving some examples of similar studies such as https://doi.org/10.3390/atmos15010039, https://doi.org/10.48129/kjs.15051 and https://doi.org/10.5109/6792841. This addition will not only contribute to a more comprehensive understanding of the subject matter but also strengthen the foundation for the subsequent research.
- L.435: “PM mass concentrations were raised at low temperatures”. Add what possible reasons for this increase.
- Low cost à low-cost: still to be corrected at different places
- L.281: The authors mentioned that the statistical indicators are presented in Table 2, however, they are not to be found in the table, please comment.
- L.414: “NA. not available” à *NA. not available, and add * in the table
- What is the reason for the low-resolution plots in Figure 17?
Author Response
Reviewer # 1
Manuscript ID: atmosphere-2851500
Title: Using a Low-cost Sensor to Estimate Fine Particulate Matter: A Case Study in Samutprakarn, Thailand
Responses to Reviewer # 1
The authors are delighted by the positive opinion of the reviewer. The reviewer mentions a number of additional minor corrections, all of which we have implemented. Please find answers to each point in the following:
Comments and Suggestions for Authors
I thank the authors for the revision. Most of my comments have been considered in the revised version. However, while reading the new version, I spotted a couple of old (not considered or justified) or new comments.
- Add a comma after equation one and start the next line with small letters.
Responses: Done.
- Add TEOM to the supplementary file.
Responses: Done.
- The authors mentioned that “PMS7003 typically has a strong correlation with the reference instrument and good reproducibility”, please elaborate on the reason for such behavior.
Responses: The authors have elaborated the reasons in the revised manuscript file, page 230-235.
- In L.305: the authors mentioned that the PM concentrations from the low-cost sensors and TEOM are shown in Figure 3, however, these figures are (not?) satisfactorily discussed in the text. Authors need to add more explanation about Figure 3, Figure 4, and Figure 5.
Responses: With thanks for the suggestion, the authors have expanded the explanations of Figures 3, 4, and 5 in the revised manuscript., page 317-350.
- The authors have successfully extended the introduction to provide additional examples and enhance the overall context of their research. However, it is noteworthy that the discussion about solar radiation is missing. Including a brief discourse on solar radiation could further enrich the background of the study. I recommend the authors to consider incorporating a few sentences on this topic by giving some examples of similar studies such as https://doi.org/10.3390/atmos15010039,
https://doi.org/10.48129/kjs.15051 and
https://doi.org/10.5109/6792841.
This addition will not only contribute to a more comprehensive understanding of the subject matter but also strengthen the foundation for the subsequent research.
Responses: The authors extended the discussion in the revised manuscript file and included 3 references as the reviewer suggestions, page 166-171.
48 Ismail, A.H.; Dawi, E.A.; Almokdad, N; Abdelkader, A; Salem, O. Estimation and Comparison of the Clearness Index using Mathematical Models-Case study in the United Arab Emirates. Atmosphere. 2023, 10 (3), 863-869. https://doi.org/10.5109/6792841.
49 Zhou, X.; Liu, S.; Shan, Y.; Endo, S.; Xie, Y.; Sengupta, M. Influences of Cloud Microphysics on the Components of Solar Irradiance in the WRF-Solar Mode. Atmosphere. 2024, 15 (1), 39. https://doi.org/10.3390/atmos15010039.
50 Abd Al Karim Haj Ismail. Prediction of global solar radiation from sunrise duration using regression functions. Kuwait J.Sci. 2022, 3: 1-8.
- 435: “PM mass concentrations were raised at low temperatures”. Add what possible reasons for this increase.
Responses: The authors explained on the reason for the changed manuscript file by the reviewer suggestions, page 486-491. Here are references.
61 Boongla, Y.; Chanonmuang, P.; Hata, M.; Furuuchi, M.; Phairuang, W. The characteristics of carbonaceous particles down to the nanoparticle range in Rangsit city in the Bangkok Metropolitan Region, Thailand. Environmental Pollution. 2021, 272, 115940.
62 Phairuang, W.; Inerb, M.; Furuuchi, M.; Hata, M.; Tekasakul, S.; Tekasakul, P. Size-fractionated carbonaceous aerosols down to PM0.1 in southern Thailand: Local and long-range transport effects. Environmental Pollution. 2020, 260, 114031.
- Low cost à??? low-cost: still to be corrected at different places
Responses: The authors corrected all errors in the revised manuscript file based on reviewer suggestions.
- 281: The authors mentioned that the statistical indicators are presented in Table 2, however, they are not to be found in the table, please comment.
Responses: The authors apologize for some mistake the statistical indicators, we have made corrections according to the reviewer’s suggestions. It presented in Table 2.
- 414: “NA. not available” à *NA. not available, and add * in the table
Responses: Done
- What is the reason for the low-resolution plots in Figure 17?
Responses: The authors have revised the previously low-resolution plot in Figure 17.
Reviewer 3 Report
Comments and Suggestions for Authors
There is still a notable absence of detailed information regarding the data processing methodologies, sensor accuracy, and indicator analysis in relation to reference stations - but not based on papers form other regions but from this particular devices used in the study. Please add on map where is TMD location. It is crucial to elucidate how the raw data were processed, how sensor accuracy was verified, and how the indicators were analyzed in comparison to the reference values obtained from these particular sensors. Without this critical information, it becomes challenging to fully assess the reliability and validity of the results presented in the study - how far away where sensors place?
Lines 170-173: This is not true! Simply put, the light scattering detector is missing small particles - therefore, it is not a problem with understanding the characteristic PM2.5 and PM1 but simply the measurement method's accuracy. Please remove or rewrite this sentence.
Again, the authors are emphasizing the spatial component of the research based only on 5 LCS. It is not enough, in my opinion.
Line 224: Is everyone affected, or just that one? Based on what is stated? This is, again, not true - each sensor should be validated separately or be in the network, which is statistically analyzed based on the reference sensors.
Fig. 17: Were the average monthly indicators calculated from raw data or after averaging in a 1-hour or 24-hour window? Does the radius of this circle indicate something? What is the purpose of showing it on the maps in that way?
I mentioned papers in the previous review; it's good that you included and mentioned them in the introduction, but as they refer to the same problem related to PM LCS and season, I would like to see more comparison and discussion of the results and methodology.
Comments on the Quality of English Languagein general is ok. just reread to add commas
Author Response
Reviewer # 3
Manuscript ID: atmosphere-2851500
Title: Using a Low-cost Sensor to Estimate Fine Particulate Matter: A Case Study in Samutprakarn, Thailand
Responses to Reviewer # 3
The authors appreciate the positive opinion and thoroughness of the reviewer. All reviewer advice has been adopted. Changes are marked throughout the revised manuscript. Please find answers for each item in the following:
Comments and Suggestions for Authors
There is still a notable absence of detailed information regarding the data processing methodologies, sensor accuracy, and indicator analysis in relation to reference stations - but not based on papers from other regions but from this particular devices used in the study. Please add on map where is TMD location. It is crucial to elucidate how the raw data were processed, how sensor accuracy was verified, and how the indicators were analyzed in comparison to the reference values obtained from these particular sensors. Without this critical information, it becomes challenging to fully assess the reliability and validity of the results presented in the study - how far away where sensors place?
Responses: The authors greatly appreciate the reviewer's constructive comments and suggestions, which have helped us to improve the manuscript. We have significantly modified and added important material in the revised manuscript. Author added the TMD location on map as presented in Fig.1. The distances between TMD and DFH1 were 9.04 km, DFH2 9.43 km, DFH3 17.04 km, and DFH4 12.89 km. The distances between 18T and DFH1 were 16.09 km, DFH2 18.89 km, DFH3 4.70 km, and DFH4 20.90 km. Additionally, the distances between 19T and DFH1 were 6.05 km, DFH2 3.72 km, DFH3 16.12 km, and DFH4 0.13 km.
Lines 170-173: This is not true! Simply put, the light scattering detector is missing small particles - therefore, it is not a problem with understanding the characteristic PM2.5 and PM1 but simply the measurement method's accuracy. Please remove or rewrite this sentence.
Responses: Thank you very much for your suggestion. The authors have removed the sentence.
Again, the authors are emphasizing the spatial component of the research based only on 5 LCS. It is not enough, in my opinion.
Responses: Thank you very much for your suggestion. The authors have removed the sentence.
Line 224: Is everyone affected, or just that one? Based on what is stated? This is, again, not true - each sensor should be validated separately or be in the network, which is statistically analyzed based on the reference sensors.
Responses: We respect the referee’s opinions but there may be a problem with the line number mentioned here, it’s hard to understand what the Referee means unfortunately. In any case, we have explained in more detail the reasons why we selected the PMS7003 sensor on Current Lines 230-235. We are pleased that the other 3 referees seem to be happy with this point.
Fig. 17: Were the average monthly indicators calculated from raw data or after averaging in a 1-hour or 24-hour window? Does the radius of this circle indicate something? What is the purpose of showing it on the maps in that way?
Responses: Thank you very much for your suggestion. We calculated from an average over 24-hours as mentioned in the paper. The reason for present figure 17 is to display the data using the colors based on the Thai AQI Index, and the co-ordinates of the PMs measuring devices were specified for display purposes. The Pollution Control Department's machine location and measurement station (two stations) supplied other comparison data. For better visual comprehension, the findings are arranged geographically by color and Thai AQI. We have also improved the resolution.
I mentioned papers in the previous review; it's good that you included and mentioned them in the introduction, but as they refer to the same problem related to PM LCS and season, I would like to see more comparison and discussion of the results and methodology.
Responses: Thank you very much for your suggestion. We have elaborated discussion of the results and methodology. The experimental data used reference stations to validate the low-cost sensor (PMS7003). The temperature and relative humidity have effects on the low-cost sensors (which have significant effect on measurements of PM concentrations), this is explained in detail in the manuscript.
Comments on the Quality of English Language
in general is ok. just reread to add commas
Responses: Thank you very much for your advice. With the help of native English speakers, particularly a visiting Professor, the authors have checked and fixed the remaining shortcomings.